# Moment Distributionally Robust Tree Structured Prediction

**Yeshu Li**      **Danyal Saeed**      **Xinhua Zhang**      **Brian D. Ziebart**
Department of Computer Science
University of Illinois at Chicago
{yli299, dsaeed3, zhangx, bziebart}@uic.edu

**Kevin Gimpel**
Toyota Technological Institute at Chicago
kgimpel@ttic.edu

## Abstract

Structured prediction of tree-shaped objects is heavily studied under the name of syntactic dependency parsing. Current practice based on maximum likelihood or margin is either agnostic to or inconsistent with the evaluation loss. Risk minimization alleviates the discrepancy between training and test objectives but typically induces a non-convex problem. These approaches adopt explicit regularization to combat overfitting without probabilistic interpretation. We propose a moment-based distributionally robust optimization approach for tree structured prediction, where the worst-case expected loss over a set of distributions within bounded moment divergence from the empirical distribution is minimized. We develop efficient algorithms for arborescences and other variants of trees. We derive Fisher consistency, convergence rates and generalization bounds for our proposed method. We evaluate its empirical effectiveness on dependency parsing benchmarks.

## 1   Introduction

Structured prediction is an important learning setting for joint prediction of interdependent variables. The output space typically consists of an exponential number of structured objects whose inherent relations can be exploited to develop efficient learning algorithms and capture key properties of data [Ciliberto et al., 2019]. Trees are widely used structures that offer expressiveness and simplicity. We distinguish between two different tree structured prediction tasks in the literature. The first task is a structure learning problem in graphical models [Bradley and Guestrin, 2010], aimed at constructing trees underlying a predictive model from training data. The optimal tree is found easily with greedy algorithms for generative models [Chow and Liu, 1968], while it is NP-hard for the discriminative max-margin setting [Meshi et al., 2013]. The second task requires prediction itself to be a tree-shaped object (e.g., an incidence vector). Dependency parsing is a crucial application of this problem that has inspired a flurry of work in natural language processing. The first-order spanning tree prediction assuming factorization over arcs can be done in $\mathcal{O}(n^2)$ [Stanojević and Cohen, 2021], whereas exact inference is NP-hard for certain (non-projective) higher-order trees (e.g., considering siblings) [McDonald and Satta, 2007]. We study the latter in this work.

A common evaluation criterion in dependency parsing is the attachment score, namely, the score we would like to maximize on test data. It is cost-sensitive to allow partially correct prediction. Ideally, the training objective should be aligned with the test objective. An early attempt to directly mimic test conditions leads to a non-convex piece-wise constant objective [Och, 2003]. Risk minimization in appropriate parametric form has a non-convex smooth objective, solvable with gradient descent,

but still losing global convergence and generalization guarantees. Maximum likelihood approaches formulate a convex smooth problem minimizing a logistic loss, consistent with conditional probability estimates but oblivious to test losses. Maximum margin methods have convex objectives able to implicitly incorporate custom losses by scaling margins, but are known to be inconsistent with test losses generally [Nowak-Vila et al., 2021]. Unfortunately, none of these approaches yield a Bayes optimal estimator for test losses with global convergence and finite-sample generalization guarantees.

Consistent structured prediction methods include Ciliberto et al. [2016], Blondel [2019], Nowak-Vila et al. [2020], the latter two of which are based on Fenchel-Young losses [Blondel et al., 2020]. However, none of them have addressed the tree structured prediction problem explicitly. For instance, Blondel [2019] calls for Euclidean or Kullback-Leibler projection oracles, which do not exist in an efficient sense from what we know for arborescence (directed tree) polytopes. In addition, the Frank-Wolfe type algorithm adopted by Nowak-Vila et al. [2020] requires a max-min oracle and converges in a rate of $\mathcal{O}(\frac{1}{\epsilon})$. Furthermore, all of the above methods belong to empirical risk minimization (ERM) that requires explicit regularization to combat overfitting, which can be quite vulnerable in high-dimensional settings (e.g., scarce data).

To address the above issues, we propose an estimator from first principles in distributionally robust optimization (DRO). It minimizes the worst-case risk over an ambiguity set of distributions within bounded moment divergence from the empirical distribution. We seek probabilistic prediction by assuming non-deterministic groundtruth labels, which, together with the ambiguity set, models uncertainty about the unknown true distribution. We interpret the primal problem as a dual-norm-regularized surrogate loss minimization problem. Note that prior art applying moment-based DRO to tree-structured graphical models [Fathony et al., 2018b] and bipartite matching [Fathony et al., 2018a] adopts a special case of our ambiguity set in which the empirical feature moments are matched exactly and regularization has to be imposed manually. This moment-based DRO also allows us to derive generalization bounds regarding true worst-case risks. When the ambiguity radius is zero, the DRO estimator is shown to be consistent. We develop two practical algorithms, one based on game theory and the other based on marginal probabilities of tree parts. We further propose efficient Euclidean projection oracles onto the arborescence polytope with linearly convergent guarantees. We conduct experiments on three common dependency parsing datasets, suggesting that our method is particularly effective with little training data.

**Contributions.** Our contributions are summarized as follows. (1) We propose a distributionally robust tree structured prediction method and show its equivalence to regularized surrogate minimization. (2) We derive its generalization bounds and consistency. (3) We propose efficient algorithms based on projection oracles for arborescence polytopes. (4) We perform empirical study on real-world datasets.

**Paper structure.** We begin with problem setup and existing work in Section 2. We present our method with theoretical analysis in Section 3. Section 4 proposes efficient projection oracles. Section 5 discusses extensions beyond first-order directed trees. Experimental results of comparing our method with a competitive baseline are given in Section 6. We conclude the paper in Section 7.

## 2 Background and Related Works

### 2.1 Tree Structured Prediction

Consider a weighted directed multi-graph $\mathcal{G} = (\mathcal{V}, \mathcal{E})$ where each arc $(i, j, l) \in \mathcal{E}$ from node $i$ to $j$ has a label $l$. By designating a root node $r \in \mathcal{V}$, we say that $\mathcal{A} \subseteq \mathcal{E}$ is an $r$-arborescence of $\mathcal{G}$ if $(\mathcal{V}, \mathcal{A})$ is a directed spanning tree rooted at $r$. For any $v \in \mathcal{V}$, denote by $\delta^-(v) := \{(i, j, l) \in \mathcal{E} : j = v\}$ the set of its incoming arcs, and $\delta^+(v) := \{(i, j, l) \in \mathcal{E} : i = v\}$ the set of its outgoing arcs.

Let $\mathcal{X}$ be the input space and $\mathcal{Y} \triangleq \bigcup_{\boldsymbol{x} \in \mathcal{X}} \mathcal{Y}(\boldsymbol{x})$ be the output space where $\mathcal{Y}(\boldsymbol{x})$ represents the set of $r$-arborescences of a graph $\mathcal{G}(\boldsymbol{x})$ formed by $\boldsymbol{x}$. Dependence on $\boldsymbol{x}$ is suppressed when context is clear. Let $\mathcal{R} \subseteq 2^{\mathcal{E}}$ be a set of parts with $\mathcal{E} \subseteq \mathcal{R}$. Each part $s \in \mathcal{R}$ is a subset of arcs. It is convenient to represent $\boldsymbol{y} \in \mathcal{Y}$ as a binary vector with $y_s = 1$ iff part $s$ appears in $\boldsymbol{y}$. Let $w_{\boldsymbol{\theta}}(\boldsymbol{x}, \boldsymbol{y}) \triangleq \sum_{s \in \mathcal{R}} w_{\boldsymbol{\theta}}(\boldsymbol{x}, y_s)$ be a score function decomposing over parts, parameterized by $\boldsymbol{\theta}$. Let $\{(\boldsymbol{x}^{(i)}, \boldsymbol{y}^{(i)})\}_{i=1}^m$ be a set of $m$ training examples drawn i.i.d. from a distribution $\mathbb{P} \in \mathcal{P}(\mathcal{X} \times \mathcal{Y})$, where each $\boldsymbol{y}^{(i)}$ is an $r$-arborescence. The goal of tree structured prediction is to learn a function $h : \mathcal{X} \to \mathcal{Y}$ from training data. Assume that the evaluation criterion is a loss function $\ell : \mathcal{Y} \times \mathcal{Y} \to \mathbb{R}_{\geq 0}$.

We introduce existing methods in the setting of (graph-based, non-projective, syntactic) dependency parsing where $\boldsymbol{x}$ is a sequence of tokens and $\mathcal{G}(\boldsymbol{x})$ encodes dependencies among tokens.

## 2.2 Maximum Likelihood

A probabilistic modeling approach based on exponential family distributions maximizes the conditional log-likelihood of the training data:

$$\min_{\boldsymbol{\theta}} - \sum_{i=1}^{m} \log p_{\boldsymbol{\theta}}(\boldsymbol{y}^{(i)}|\boldsymbol{x}^{(i)}) := - \sum_{i=1}^{m} \log \left[ \exp \left( w_{\boldsymbol{\theta}}(\boldsymbol{x}^{(i)}, \boldsymbol{y}^{(i)}) \right) / Z(\boldsymbol{x}^{(i)}) \right],$$

where $Z(\boldsymbol{x}) \triangleq \sum_{\boldsymbol{y} \in \mathcal{Y}(\boldsymbol{x})} \exp \left( w_{\boldsymbol{\theta}}(\boldsymbol{x}, \boldsymbol{y}) \right)$. This problem is convex for log-linear models, but intractable for general $\mathcal{R}$ [Koller and Friedman, 2009]. The first-order arc-factored model ($\mathcal{R} = \mathcal{E}$) is equivalent to a loop-free factor graph, rendering it tractable via the matrix-tree theorem [Kirchhoff, 1847, William, 1984, Koo et al., 2007, McDonald and Satta, 2007, Smith and Smith, 2007]. Neural parsers either leverage the same theorem to compute the partition function [Ma and Hovy, 2017] or consider the parent node distribution independently for each node by local normalization [Dozat and Manning, 2017, Zhang et al., 2017]. Higher-order models require approximate algorithms such as loopy belief propagation [Murphy et al., 1999] and Markov chain Monte Carlo [Brooks, 1998]. This approach does not incorporate task-specific losses. In fact, with maximum a posteriori (MAP) decoding, it is not consistent with any specific loss in general [Nowak-Vila et al., 2019].

## 2.3 Maximum Margin

An alternative approach based on maximum margin Markov networks [Taskar et al., 2003] or structured support vector machines [Tsochantaridis et al., 2005] optimizes a hinge-type surrogate:

$$\min_{\boldsymbol{\theta}} \sum_{i=1}^{m} -w_{\boldsymbol{\theta}}(\boldsymbol{x}^{(i)}, \boldsymbol{y}^{(i)}) + \max_{\boldsymbol{y}} \ell(\boldsymbol{y}^{(i)}, \boldsymbol{y}) + w_{\boldsymbol{\theta}}(\boldsymbol{x}^{(i)}, \boldsymbol{y}),$$

which inspires a rich line of work based on MAP inference with manual features [Taskar et al., 2004, McDonald et al., 2005, McDonald and Pereira, 2006, Martins et al., 2009, 2010, 2015, Zhang et al., 2014] or deep learning [Kiperwasser and Goldberg, 2016, Wang and Chang, 2016]. Approximate MAP inference is required for models beyond first-order. A smooth variant called softmax-margin [Gimpel and Smith, 2010] incorporates the task-specific loss $\ell$ but still implicitly minimizes it. Margin-based objectives are known to be consistent only under very restrictive conditions [Liu, 2007, Nowak-Vila et al., 2021] (i.e., data with majority label, loss being a distance).

## 2.4 Minimum Risk

Empirical risk minimization suggests directly optimizing the expected target loss on training data:

$$\min_{\boldsymbol{\theta}} \sum_{i=1}^{m} \sum_{\boldsymbol{y}} p_{\boldsymbol{\theta}}(\boldsymbol{y}|\boldsymbol{x}^{(i)}) \ell(\boldsymbol{y}^{(i)}, \boldsymbol{y}),$$

which is commonly non-convex due to normalization of $p_{\boldsymbol{\theta}}$. There are a few parsers optimizing this objective via back-propagation [Stoyanov and Eisner, 2012], $k$-best lists [Smith and Eisner, 2006], semirings [Li and Eisner, 2009, Zmigrod et al., 2021] and other differentiable approximations [Gormley et al., 2015, Mensch and Blondel, 2018]. Local optima found by these algorithms do not satisfy the premise of Fisher consistency and make it difficult to quantify generalization errors.

## 2.5 Distributionally Robust Optimization

Distributionally robust optimization has attracted emerging interests in improving machine learning models due to its connections to robustness, regularization and generalization. It proposes to minimize a risk with respect to the worst-case distribution chosen by an adversary in some uncertainty set:

$$\min_{\boldsymbol{\theta}} \max_{\mathbb{Q} \in \mathcal{B}} \mathbb{E}_{\mathbb{Q}}[\ell(\boldsymbol{Y}, h_{\boldsymbol{\theta}}(\boldsymbol{X}))],$$

where $\mathcal{B}$ is an ambiguity set that can be defined by discrepancies [Shafieezadeh-Abadeh et al., 2019, Duchi and Namkoong, 2019], moments [Delage and Ye, 2010, Farnia and Tse, 2016], shapes [Popescu, 2005, Hanasusanto et al., 2015] and kernels [Shang et al., 2017, Staib and Jegelka, 2019]. A thorough review can be found in Rahimian and Mehrotra [2019]. We focus on moment-matching discriminative approaches while a similar generative method is proposed in Ganapathi et al. [2008].

## 3 Method

We introduce the formulation, followed by practical algorithms for learning and inference. Afterwards, we present the theoretical guarantees. We defer all proofs to Appendix A.

### 3.1 Formulation

We assume that the evaluation criterion is the Hamming loss $\ell(\boldsymbol{y}, \boldsymbol{y}') := \sum_i \mathbb{1}(y_i \neq y_i')$ with $\mathbb{1}(\cdot)$ being the 0-1 indicator function, but the results in this paper generalize to losses with affine decomposition [Ramaswamy et al., 2013] easily.

Let $\mathbb{P}^{\text{true}}$ be the true distribution and $\mathbb{P}^{\text{emp}}$ be the empirical distribution. Our approach builds upon a probabilistic predictor that non-parametrically minimizes the expected loss with regard to the most adverse distribution in an uncertainty set where the distributions are $\varepsilon$ away from the empirical distribution in terms of feature moment difference:

$$\min_{\mathbb{P}} \max_{\mathbb{Q} \in \mathcal{B}(\mathbb{P}^{\text{emp}})} \mathbb{E}_{\mathbb{Q}_{\boldsymbol{X}, \check{\boldsymbol{Y}}}, \mathbb{P}_{\hat{\boldsymbol{Y}} | \boldsymbol{X}}} \ell(\hat{\boldsymbol{Y}}, \check{\boldsymbol{Y}}), \tag{1}$$

where $\mathcal{B}(\mathbb{P}^{\text{emp}}) := \{\mathbb{Q} : \mathbb{Q}_{\boldsymbol{X}} = \mathbb{P}_{\boldsymbol{X}}^{\text{emp}} \land \|\mathbb{E}_{\mathbb{P}^{\text{emp}}} \boldsymbol{\phi}(\cdot) - \mathbb{E}_{\mathbb{Q}} \boldsymbol{\phi}(\cdot)\| \leq \varepsilon\}$ with $\varepsilon \geq 0$ and $\boldsymbol{\phi} : \mathcal{X} \times \mathcal{Y} \to \mathbb{R}^d$ is a joint feature mapping decomposable over parts: $\boldsymbol{\phi}(\boldsymbol{x}, \boldsymbol{y}) \triangleq \sum_s \boldsymbol{\phi}(\boldsymbol{x}, y_s)$. In Farnia and Tse [2016], cross-moments are adopted: $\boldsymbol{\phi}(\boldsymbol{x}, \boldsymbol{y}) := \boldsymbol{\phi}_{\boldsymbol{X}}(\boldsymbol{x}) \otimes \boldsymbol{\phi}_{\boldsymbol{Y}}(\boldsymbol{y})$ where $\otimes$ is the tensor product.

By Fenchel duality [Altun and Smola, 2006] and strong duality [Von Neumann and Morgenstern, 1947], we show that Eq. (1) is analogous to dual-norm-regularized surrogate loss minimization:

**Proposition 1.** *The distributionally robust tree structured prediction problem based on moment divergence in Eq.* (1) *can be rewritten as*

$$\min_{\boldsymbol{\theta}} \mathbb{E}_{\mathbb{P}_{\boldsymbol{X}, \boldsymbol{Y}}^{emp}} \underbrace{\min_{\mathbb{P}} \max_{\mathbb{Q}} \mathbb{E}_{\mathbb{P}_{\hat{\boldsymbol{Y}} | \boldsymbol{X}}, \mathbb{Q}_{\check{\boldsymbol{Y}} | \boldsymbol{X}}} \ell(\hat{\boldsymbol{Y}}, \check{\boldsymbol{Y}}) + \boldsymbol{\theta}^{\mathsf{T}}(\boldsymbol{\phi}(\boldsymbol{X}, \check{\boldsymbol{Y}}) - \boldsymbol{\phi}(\boldsymbol{X}, \boldsymbol{Y})) + \varepsilon \|\boldsymbol{\theta}\|_*}_{\ell_{adv}(\boldsymbol{\theta}, (\boldsymbol{X}, \boldsymbol{Y}))}, \tag{2}$$

*where $\boldsymbol{\theta} \in \mathbb{R}^d$ is the vector of Lagrangian multipliers and $\|\cdot\|_*$ is the dual norm of $\|\cdot\|$.*

### 3.2 Constraint Generation Solution

From a game-theoretic rationale [Topsøe, 1979, Grünwald and Dawid, 2004], Eq. (1) is considered as an adversary-constrained zero-sum game. A prediction player chooses a set of stochastic strategies (conditional distributions over arborescences) in order to minimize the expected payoff whereas an adversarial player chooses constrained strategies to maximize it. The payoff for a pair of pure strategies is the incurred loss, $\ell(\hat{\boldsymbol{y}}, \check{\boldsymbol{y}})$. The constrained game is transformed to a set of unconstrained ones in Eq. (2) whose payoffs are parameterized by $\boldsymbol{\theta}$: $\text{payoff}(\hat{\boldsymbol{y}}, \check{\boldsymbol{y}}) \triangleq \ell(\hat{\boldsymbol{y}}, \check{\boldsymbol{y}}) + \boldsymbol{\theta}^{\mathsf{T}} \boldsymbol{\phi}(\boldsymbol{x}, \check{\boldsymbol{y}})$. Note that the games in Eq. (1) are jointly constrained for all $\boldsymbol{x}$'s in the support of $\mathbb{P}_{\boldsymbol{X}}^{\text{emp}}$ while the ones in Eq. (2) are conditionally independent given $\boldsymbol{x}$. The unconstrained game can be solved by a linear program [Von Neumann and Morgenstern, 1947]. However, there are $\mathcal{O}(n^n)$ spanning trees in a complete graph, thus making explicit construction of the full payoff matrix impractical.

We adopt a constraint generation algorithm named double oracle [McMahan et al., 2003], shown in Appendix B. It builds a payoff sub-matrix starting from small initial sets of strategies. In each iteration, each player takes their turn based on the game payoff sub-matrix by finding the best response among all possible strategies to the opponent's optimal mixture strategies. The response is added to a player's strategy set if it improves the value of the game, with the sub-matrix updated. The algorithm terminates and converges to a Nash equilibrium of the original game when the strategy sets no longer grow. The size of the final sub-matrix is usually small in practice but there are no known theoretical guarantees, thus no way to analyze the convergence behavior. Finding the best response requires an

oracle, equivalent to finding the minimum weight arborescence. The objective in Eq. (2) is a convex function of $\boldsymbol{\theta}$, so we can optimize it with sub-gradients based on solutions of the inner zero-sum games. Although lacking convergence guarantees, this algorithm is flexible with custom losses and provides a game-theoretic perspective to a typical DRO problem.

## 3.3 Marginal Distribution Formulation

The $r$-arborescence polytope is defined as the convex hull of all vectors representing $r$-arborescences: $\mathcal{A}_{\text{arb}}(\boldsymbol{x}) := \text{Conv}(\{\boldsymbol{y} \in \mathbb{R}^{|\mathcal{R}|} : \boldsymbol{y} \in \mathcal{Y}(\boldsymbol{x})\})$. Note that each $\boldsymbol{p} \in \mathcal{A}_{\text{arb}}$ is a convex combination of all $r$-arborescences: $\boldsymbol{p} \triangleq \sum_{\boldsymbol{y}} \text{Prob}(\boldsymbol{y})\boldsymbol{y}$, where $p_s$ denotes the marginal probability of part $s$. Here we adopt the squared $\ell_2$ norm as the dual norm and an ambiguity radius of $\varepsilon = \lambda/2$. By substituting the marginal probability vectors and switching min-max optimization orders, we simplify Eq. (2) into

$$\max_{\boldsymbol{q}^{(i)} \in \mathcal{A}_{\text{arb}}} \min_{\boldsymbol{\theta}} \frac{1}{m} \sum_{i=1}^{m} \min_{\boldsymbol{p} \in \mathcal{A}_{\text{arb}}} (\boldsymbol{q}^{(i)} - \boldsymbol{p}_{\text{emp}}^{(i)})^{\mathsf{T}} \boldsymbol{\Phi}^{(i)} \boldsymbol{\theta} - \langle \boldsymbol{p}, \boldsymbol{q}^{(i)} \rangle + \frac{\mu}{2}\|\boldsymbol{p}\|_2^2 - \frac{\mu}{2}\|\boldsymbol{q}^{(i)}\|_2^2 + \frac{\lambda}{2}\|\boldsymbol{\theta}\|_2^2, \quad (3)$$

where $\boldsymbol{\Phi}^{(i)} \in \mathbb{R}^{|\mathcal{R}| \times d}$ denotes the feature matrix of the $i$-th training data, $\mu \in \mathbb{R}_{\geq 0}$ is a smoothing parameter to induce strong convexity. We push the maximization over $\boldsymbol{q}$ to the outermost level because of its large computational cost. If $\mu = 0$, the solution to Eq. (3) is also optimal to Eq. (2) by strong duality but the problem becomes non-smooth. Therefore we expect $\boldsymbol{\theta}^*$ obtained with a very small positive $\mu$ to be a good approximation of $\boldsymbol{\theta}^*$ obtained with $\mu = 0$.

To optimize it, with fixed $\boldsymbol{q}$, due to strong convexity, the unconstrained minimization over $\boldsymbol{\theta}$ yields $\boldsymbol{\theta}^* = -\frac{1}{m\lambda} \sum_{i=1}^{m} (\boldsymbol{\Phi}^{(i)})^{\mathsf{T}} (\boldsymbol{q}^{(i)} - \boldsymbol{p}_{\text{emp}}^{(i)})$. In contrast, the constrained minimization over $\boldsymbol{p}$ admits no closed-form solution but can be cast as Euclidean projection onto $\mathcal{A}_{\text{arb}}$ instead, independently for each $i \in [m]$: $\boldsymbol{p}^* = \min_{\boldsymbol{p} \in \mathcal{A}_{\text{arb}}} \|\boldsymbol{p} - \frac{1}{\mu}\boldsymbol{q}^{(i)}\|_2^2 \triangleq \text{Proj}_{\mathcal{A}_{\text{arb}}}(\frac{1}{\mu}\boldsymbol{q}^{(i)})$. Given $\boldsymbol{\theta}^*$ and $\boldsymbol{p}^*$, the outermost maximization can be solved by a projected quasi-Newton algorithm [Schmidt et al., 2009] that also requires the projection oracle $\text{Proj}_{\mathcal{A}_{\text{arb}}}(\cdot)$, elaborated in Section 4.

## 3.4 Inference

We propose two algorithms to make inference with given $\boldsymbol{\theta}^*$.

**Weight construction.** Construct the part weights as $\boldsymbol{\Phi}\boldsymbol{\theta}^* \in \mathbb{R}^{|\mathcal{R}|}$ and find the maximum weight arborescence: $\boldsymbol{y}^* \in \arg\max_{\boldsymbol{y}} \boldsymbol{y}^{\mathsf{T}} \boldsymbol{\Phi}\boldsymbol{\theta}^*$ by the Gabow-Tarjan (GT) algorithm [Gabow et al., 1986, Zmigrod et al., 2020] or approximate methods for higher-order trees.

**Minimum Bayes risk decoding.** The optimal probabilistic prediction $\mathbb{P}^*$ or $\boldsymbol{p}^*$ can be obtained from Eq. (2) or Eq. (3). The marginal probabilities enable minimum Bayes risk decoding: $\boldsymbol{y}^* \in \arg\min_{\boldsymbol{y}} \mathbb{E}_{\mathbb{P}^*_{\hat{\boldsymbol{Y}}|\boldsymbol{x}}} \ell(\boldsymbol{y}, \hat{\boldsymbol{Y}}) \triangleq \arg\max_{\boldsymbol{y}} \sum_{s:y_s=1} \boldsymbol{p}_s^*$, a maximum weight arborescence problem.

## 3.5 Statistical Properties

Basic generalization bounds of DRO methods derived from measure concentration are not appropriate for an ambiguity set defined by low-order moments in this paper since it fails to converge [Shafieezadeh-Abadeh et al., 2019]. We take an alternate approach following Farnia and Tse [2016] to obtain excess out-of-sample risk bounds by assuming boundedness on features and losses.

**Theorem 2.** *Given $m$ samples, a non-negative loss $\ell(\cdot, \cdot)$ such that $|\ell(\cdot, \cdot)| \leq K$, a feature function $\phi(\cdot, \cdot)$ such that $\|\phi(\cdot, \cdot)\| \leq B$, a positive ambiguity level $\varepsilon > 0$, then, for any $\rho \in (0, 1]$, with a probability at least $1 - \rho$, the following excess true worst-case risk bound holds:*

$$\max_{\mathbb{Q} \in \mathcal{B}(\mathbb{P}^{true})} R_{\mathbb{Q}}^L(\boldsymbol{\theta}_{emp}^*) - \max_{\mathbb{Q} \in \mathcal{B}(\mathbb{P}^{true})} R_{\mathbb{Q}}^L(\boldsymbol{\theta}_{true}^*) \leq \frac{4KB}{\varepsilon\sqrt{m}} \left(1 + \frac{3}{2}\sqrt{\frac{\ln(4/\rho)}{2}}\right),$$

*where $\boldsymbol{\theta}_{emp}^*$ and $\boldsymbol{\theta}_{true}^*$ are the optimal parameters learned in Eq. (2) under $\mathbb{P}^{emp}$ and $\mathbb{P}^{true}$ respectively. The original risk of $\boldsymbol{\theta}$ under $\mathbb{Q}$ is $R_{\mathbb{Q}}^L(\boldsymbol{\theta}) := \mathbb{E}_{\mathbb{Q}_{\boldsymbol{X},\boldsymbol{Y}}, \mathbb{P}_{\hat{\boldsymbol{Y}}|\boldsymbol{X}}^{\boldsymbol{\theta}}} \ell(\hat{\boldsymbol{Y}}, \boldsymbol{Y})$ with Bayes prediction $\mathbb{P}_{\boldsymbol{Y}|\boldsymbol{x}}^{\boldsymbol{\theta}} \in \arg\min_{\mathbb{P}} \max_{\mathbb{Q}} \mathbb{E}_{\mathbb{Q}_{\check{\boldsymbol{Y}}|\boldsymbol{x}} \mathbb{P}_{\hat{\boldsymbol{Y}}|\boldsymbol{x}}} \ell(\hat{\boldsymbol{Y}}, \check{\boldsymbol{Y}}) + \boldsymbol{\theta}^{\mathsf{T}} \phi(\boldsymbol{x}, \check{\boldsymbol{Y}})$.*

Theorem 2 presents a bound based on uniform convergence and Rademacher complexities [Bartlett and Mendelson, 2002], which improves the results in Asif et al. [2015], who merely show that the worst-case risk upper bounds the risk under any distribution in the ambiguity set.

The dual problem in Eq. (2) suggests an adversarial surrogate loss $\ell_{\mathrm{adv}}(\boldsymbol{\theta}, (\boldsymbol{x}, \boldsymbol{y}))$ in a ERM form. The special case of $\varepsilon = 0$ in our DRO estimator has a similar form to the max-min surrogate loss in Nowak-Vila et al. [2020] except that we assume probabilistic prediction. A conclusion of its Fisher consistency can thus be drawn based on Fathony et al. [2018a], Nowak-Vila et al. [2020].

**Corollary 3.** *When $\varepsilon = 0$, $\ell_{adv}$ is Fisher consistent with respect to $\ell$. Namely, $\mathbb{P}_{\hat{\boldsymbol{Y}}|\boldsymbol{X}}^{\boldsymbol{\theta}_{true}^*}$ is the probabilistic prediction made by the Bayes optimal decision rule, where $\boldsymbol{\theta}_{true}^*$ is defined in Theorem 2.*

If $\varepsilon > 0$, the decoded prediction for each $\boldsymbol{x}$ will not belong to the convex hull of true conditional distributions, thus not a minimizer of $\ell$. On the other hand, if $\varepsilon$ is chosen as $m^{-\alpha}$ for $0 < \alpha < 1/2$, $\ell_{\mathrm{adv}}$ will be universally consistent according to the comparison inequality in Nowak-Vila et al. [2020].

# 4 Projection onto Arborescence Polytopes

The Euclidean projection onto an $r$-arborescence polytope is a quadratic programming problem[1]:

$$\min_{\boldsymbol{x} \in \mathcal{A}_{\mathrm{arb}}} f(\boldsymbol{x}) := \|\boldsymbol{x} - \boldsymbol{w}\|_2^2.$$

We focus on first-order models and discuss the extensions to other classes of trees in Section 5.

## 4.1 Frank-Wolfe Algorithm

The Frank-Wolfe (FW) method [Frank et al., 1956] is an iterative first-order algorithm that enforces constraints by optimizing a linear objective over the feasible set at each iteration $t$:

$$\boldsymbol{s}^t \in \arg \min_{\boldsymbol{s} \in \mathcal{A}_{\mathrm{arb}}} \boldsymbol{s}^\mathsf{T} \nabla f(\boldsymbol{x}^t), \tag{4}$$

which is a minimum weight arborescence problem with weights $\nabla f(\boldsymbol{x}^t)$ in our case. The solution is updated and stays feasible: $\boldsymbol{x}^{t+1} \leftarrow \boldsymbol{x}^t + \gamma_t(\boldsymbol{s}^t - \boldsymbol{x}^t)$, where $\gamma_t$ is a step size typically set to $\frac{2}{t+2}$. FW style algorithms are known to have a convergence rate of $\mathcal{O}(\frac{1}{\epsilon})$ [Jaggi, 2013].

## 4.2 Martin's Polytope

A compact representation of $\mathcal{A}_{\mathrm{arb}}$ with a polynomial number of linear constraints is attractive to lead to efficient algorithms. To the best of our knowledge, there is no existing projection method exploiting special structures of this polytope. An extended formulation of the arborescence polytope [Friesen, 2019, Martin, 1991] follows a lift-and-project approach. It relates each element to existence of $k$-arboresences of the underlying undirected graph for all $k \in \mathcal{V}$. We extend it to multi-graphs:

$$\mathcal{A}_{\mathrm{marb}} := \{\boldsymbol{z}^r : \exists \boldsymbol{z}^k \geq \boldsymbol{0} \sum_{a \in \delta^-(j)} z_a^k = \mathbb{1}(j \neq k) \, \forall k, j \in \mathcal{V} \wedge \sum_{a \in \mathcal{E}'_{ij}} z_a^k = \sum_{a \in \mathcal{E}_{ij}} z_a^r \, \forall k \neq r, i, j \in \mathcal{V} \wedge \boldsymbol{z}^r \geq \boldsymbol{0}\},$$

where $\boldsymbol{z}^r \in \mathbb{R}^{|\mathcal{E}|}$ is associated with the original arcs $\mathcal{E}$, $\boldsymbol{z}^k \in \mathbb{R}^{|\mathcal{E}'|}$ for $k \neq r$ is associated with a simple directed graph $(\mathcal{V}, \mathcal{E}')$ formed by removing directions and splitting each edge $\{i, j\}$ into two directed ones, $\mathcal{E}_{ij} := \{a \in \mathcal{E} : \bar{a} = \{i, j\}\}$ is the set of arcs connecting $i$ and $j$ with $\bar{a} \triangleq \overline{(i, j, l)} := \{i, j\}$ denoting the underlying undirected edge. We show exact correspondence between $\mathcal{A}_{\mathrm{marb}}$ and $\mathcal{A}_{\mathrm{arb}}$ based on a similar argument for simple graphs [Friesen, 2019]:

**Proposition 4.** *Let $\mathcal{G}$ be a multi-graph. $\mathcal{A}_{marb} \triangleq \mathcal{A}_{arb}$.*

---

[1]This is a well-defined convex optimization problem, different from that in differentiable structured prediction methods [Peng et al., 2018, Mihaylova et al., 2020] which elicit gradients with respect to inputs.

To solve $\min_{\boldsymbol{x} \in \mathcal{A}_{\text{marb}}} \|\boldsymbol{x} - \boldsymbol{w}\|_2^2$, we propose to adopt the alternating direction method of multipliers (ADMM) and rewrite it into the following separable form:

$$\min_{\boldsymbol{u}} g(\boldsymbol{u}) := \sum_{k \in \mathcal{V}} \frac{1}{|\mathcal{V}|} \|\boldsymbol{u}_k - \boldsymbol{w}\|_2^2 + I_{\mathcal{U}_k}(\boldsymbol{u}_k)$$

$$\text{s.t.} \quad \mathcal{U}_k := \{\boldsymbol{x} \in \mathbb{R}^{|\mathcal{E}|} : \exists \boldsymbol{z} \in \mathbb{R}_{\geq 0}^{|\mathcal{E}'|} \sum_{a \in \delta^-(j)} z_a = \mathbb{1}(j \neq k) \wedge \sum_{a \in \mathcal{E}_{ij}'} z_a = \sum_{a \in \mathcal{E}_{ij}} x_a \; \forall i, j \in \mathcal{V}\}$$

$$\boldsymbol{u}_r = \boldsymbol{u}_k \quad \forall k \in \mathcal{V} \setminus r, \quad \mathcal{U}_r := \{\boldsymbol{x} \in \mathbb{R}_{\geq 0}^{|\mathcal{E}|} : \sum_{a \in \delta^-(j)} x_a = \mathbb{1}(j \neq r) \; \forall j \in \mathcal{V}\},$$

where $I_{\mathcal{U}}(\cdot)$ is the characteristic function with $I_{\mathcal{U}}(\boldsymbol{x}) = 0$ if $\boldsymbol{x} \in \mathcal{U}$ and $\infty$ otherwise.

Let $\boldsymbol{\lambda}_k'$ be the dual variables and $\boldsymbol{\lambda}_k := \frac{1}{\rho_k} \boldsymbol{\lambda}_k'$. The scaled augmented Lagrangian function is $L_\rho(\boldsymbol{u}, \boldsymbol{\lambda}) = g(\boldsymbol{u}) + \sum_{k \neq r} \frac{\rho_k}{2} \|\boldsymbol{u}_r - \boldsymbol{u}_k + \boldsymbol{\lambda}_k\|_2^2 - \frac{\rho_k}{2} \|\boldsymbol{\lambda}_k\|_2^2$.

The ADMM algorithm updates the parameters as follows:

$$\boldsymbol{u}_k^{t+1} := \arg \min_{\boldsymbol{u}_k \in \mathcal{U}_k} L_\rho((\boldsymbol{u}_r^t, \boldsymbol{u}_k^t), \boldsymbol{\lambda}^t) \triangleq \text{Proj}_{\mathcal{U}_k}\left(\frac{2\boldsymbol{w} + \rho_k |\mathcal{V}|(\boldsymbol{u}_r^t + \boldsymbol{\lambda}_k^t)}{2 + \rho_k |\mathcal{V}|}\right) \quad \forall k \neq r$$

$$\boldsymbol{u}_r^{t+1} := \arg \min_{\boldsymbol{u}_r \in \mathcal{U}_r} L_\rho((\boldsymbol{u}_r^t, \boldsymbol{u}_k^{t+1}), \boldsymbol{\lambda}^t) \triangleq \text{Proj}_{\mathcal{U}_r}\left(\frac{2\boldsymbol{w} + |\mathcal{V}| \sum_{k \neq r} \rho_k (\boldsymbol{u}_k^{t+1} - \boldsymbol{\lambda}_k^t)}{2 + |\mathcal{V}| \sum_{k \neq r} \rho_k}\right)$$

$$\boldsymbol{\lambda}_k^{t+1} := \boldsymbol{\lambda}_k^t + (\boldsymbol{u}_r^{t+1} - \boldsymbol{u}_k^{t+1}) \quad \forall k \neq r.$$

This decomposes the original projection problem into simpler projection problems. Projection onto $\mathcal{U}_k$ for $k = r$ decomposes over $j \in \mathcal{V}$ into $|\mathcal{V}|$ projections onto simplex, solvable as fast as $\mathcal{O}(n)$ in the worst case [Condat, 2016]. For $k \neq r$, computation of $\boldsymbol{u}_k^{t+1}$ can be done in parallel. The Lagrange dual problem of $\text{Proj}_{\mathcal{U}_k}(\cdot)$ can be written as

$$\max_{\boldsymbol{\alpha} \in \mathbb{R}^{|\mathcal{V}|}} \sum_{\{i,j\} \in \bar{\mathcal{E}}} h_{ij}(\boldsymbol{\alpha}) - \sum_{j \neq k} \alpha_j \quad \text{s.t.} \quad h_{ij}(\boldsymbol{\alpha}) = \begin{cases} w_{ij}^2 / n_{ij} & \text{if } \alpha_{ij} > 2w_{ij}/n_{ij}, \\ -n_{ij}\alpha_{ij}^2/4 + \alpha_{ij}w_{ij} & \text{if } \alpha_{ij} \leq 2w_{ij}/n_{ij}, \end{cases}$$

where $w_{ij} := \sum_{a \in \mathcal{E}_{ij}} w_a$, $n_{ij} := |\mathcal{E}_{ij}|$, $\alpha_{ij} := \min(\alpha_i, \alpha_j)$ and $\alpha_k := +\infty$. Strong duality holds by linear constraint qualification. Primal solutions are recovered by $x_a^* = w_a - \min(\alpha_{\bar{a}}^*/2, w_{\bar{a}}/n_{\bar{a}})$.

**Convergence.** The dual objective of $\text{Proj}_{\mathcal{U}_k}(\cdot)$ is strongly concave on $\{\boldsymbol{\alpha} \in \mathbb{R}^{|\mathcal{V}|} : \forall i \exists j \; \{i,j\} \in \bar{\mathcal{E}} \wedge \alpha_i \leq \alpha_j \wedge \alpha_i \leq 2w_{ij}/n_{ij}\}$, with a unique global maximizer. This implies fast convergence in practice given good initialization. The negative Lagrange dual function has restricted strong convexity with $\nu = \min_{ij}(n_{ij}/2)$, near the optimum, suggesting linear convergence [Zhang and Cheng, 2015]. Alternatively, exact solutions can be found by enumerating rankings (with duplicates) of $\boldsymbol{\alpha}$ in $\mathcal{O}(|\mathcal{V}|^{|\mathcal{V}|})$. In this manner, the ADMM algorithm with a strongly convex objective has a linear convergence rate $\mathcal{O}(\log \frac{1}{\epsilon})$ with either exact [Deng and Yin, 2016] or linearly convergent approximate solution [Hager and Zhang, 2020] of $\text{Proj}_{\mathcal{U}_k}(\cdot)$. Using Nesterov's accelerated gradient algorithm [Nesterov, 2003] to optimize Eq. (3) leads to iteration complexity $\mathcal{O}(C \log \frac{1}{\epsilon})$ with constant $C$ dependent on Lipschitz constants of gradients and $\mu$.

## 5 Extensions

### 5.1 Undirected Spanning Trees

An straight-forward way of extending to undirected spanning trees is to split $\{i, j\}$ into two arcs $(i, j), (j, i)$ and make the feature mapping direction-invariant, i.e., $\boldsymbol{\phi}(\boldsymbol{x}, y_s) = \boldsymbol{\phi}(\boldsymbol{x}, y_{s'})$ for $s$ and $s'$ having the same underlying undirected graph. We post-process the prediction by removing directions.

Alternatively, we seek projection oracles for undirected graphs. Projection via FW is done by using any minimum spanning tree algorithm in Eq. (4). For ADMM, the formulation in Martin [1991] is originally for undirected trees: $\mathcal{A}_{\text{mund}} := \{\boldsymbol{x} : \exists \boldsymbol{z} \geq \boldsymbol{0} \sum_{a \in \delta^-(j)} z_a^k = \mathbb{1}(j \neq k) \wedge z_{ij}^k + z_{ji}^k = x_{\{i,j\}} \forall k, i, j \in \mathcal{V}\}$. ADMM is easily adapted to this case with $\sum_{a \in \mathcal{E}_{ij}} x_a$ replaced by $x_{\{i,j\}}$.

## 5.2 Dependency Trees

The spanning tree structure in dependency parsing is a special one where the outdegree of root is restricted to be one. We can use the GT algorithm for inference with either the same training objective or an aligned objective where a dependency tree polytope is considered: $\mathcal{A}_{\text{dep}}(\boldsymbol{x}) := \text{Conv}(\{\boldsymbol{y} \in \mathcal{Y}(\boldsymbol{x}) : |\delta^+(r)| = 1\})$. A straightforward extension of $\mathcal{A}_{\text{marb}}$ to characterizing dependency trees is $\mathcal{A}_{\text{mdep}} := \{\boldsymbol{z}^r : \boldsymbol{z}^r \in \mathcal{A}_{\text{marb}} \wedge \sum_{a \in \delta^+(r)} z_a^r = 1\}$, equivalent to $\mathcal{A}_{\text{dep}}$ by the following proposition:

**Proposition 5.** *Let $\mathcal{G}$ be a multi-graph. $\mathcal{A}_{mdep} \triangleq \mathcal{A}_{dep}$.*

FW methods leverage the GT algorithm in Eq. (4). As for ADMM, the dual problem of projection onto $\mathcal{U}_r' := \{\boldsymbol{x} : \boldsymbol{x} \in \mathcal{U}_r \wedge \sum_{a \in \delta^+(r)} x_a = 1\}$ becomes

$$\max_{\boldsymbol{\alpha},\beta} \sum_{a \in \mathcal{E}} h_a(\boldsymbol{\alpha}, \beta) - \sum_{j \neq r} \alpha_j - \beta \quad \text{s.t. } h_a(\boldsymbol{\alpha}, \beta) = \begin{cases} w_a^2 & \gamma_a > 2w_a, \\ w_a \gamma_a - \gamma_a^2/4 & \gamma_a \leq 2w_a, \end{cases}$$

where $\gamma_{(i,j,l)} := \alpha_j + \mathbb{1}(i = r)\beta$. This can be solved in $\mathcal{O}(|\mathcal{E}| \log |\mathcal{E}|)$ [Zhang et al., 2010].

## 5.3 Higher-order Polytope

Compact higher-order polytope descriptions exist for undirected spanning trees but are still unknown for arborescences with even one monomial [Friesen, 2019]. FW requires a linear oracle that is NP-hard to solve exactly in higher-order settings [McDonald and Pereira, 2006].

Instead, we can approximate it with a local polytope where the marginal probabilities of each part $s$ is required to be locally consistent with that of each arc $a$. For simplicity, we consider only features for the all-true assignments, i.e., all arcs exist in part $s$. The resulting polytope can be written as $\mathcal{A}_{\text{mloc}} := \{\boldsymbol{x} : \boldsymbol{x}_{\mathcal{E}} \in \mathcal{A}_{\text{marb}} \wedge \forall s \in \mathcal{R}, a \in s \quad p_s \leq p_a\}$, which suggests an ADMM algorithm with additional constraint sets for each part: $\mathcal{U}_s := \{\boldsymbol{x} \in \mathbb{R}_{\geq 0}^{|\mathcal{R}|} : x_s \leq x_a \quad \forall a \in s\}$, the projection onto which can be done in $\mathcal{O}(|s| \log |s|)$. See Appendix D for details.

# 6 Experiments

We evaluate our proposed method on dependency parsing tasks and compare its ability to *BiAF* [Dozat and Manning, 2017], arguably the state-of-the-art neural dependency parser. We implement our methods in Python and C[2]. We leverage the implementations in SuPar[3] [Zhang et al., 2020] for the baseline. All experiments are conducted on a computer with an Intel Core i7 CPU (2.7 GHz) and an NVIDIA Tesla P100 GPU (16 GB).

We adopt three public datasets, the English Penn Treebank (PTB v3.0) [Marcus et al., 1993], the Penn Chinese Treebank (CTB v5.1) [Xue et al., 2002] and the Universal Dependencies (UD v2.3) [Nivre et al., 2016]. See Appendix C for data-processing details.

Representation learning is not the focus of this paper. We follow Levy et al. [2020] and compare our method with the last biaffine classification layer in *BiAF* on top of pretrained features preceding this layer (backbone's output). The pretrained embeddings produced by complicated non-linear models make Fisher consistency's assumption of optimizing over all measurable functions less violated. To featurize the data, for each dataset, we train a *BiAF* network with the whole training set to obtain a pretrained model. Note that this may create unfair advantages for the baseline because the last layer was optimized together with the backbone network in an end-to-end manner during pretraining. Moreover, pretraining uses a standard ERM objective with the cross-entropy loss and local normalization over head nodes. The pretrained features are thus more adequate for the ERM objective than for our DRO objective. To make use of the features as inputs in our method, we take the outer product of the embedding vectors for two nodes as the arc feature vector. Our method and the biaffine layer therefore share the same number of parameters ($501 \times 501$, including bias terms). We focus on predicting the unlabeled dependency tree while relying on pretrained models for relation label prediction. The evaluation criteria are the labeled/unlabeled attachment scores (LAS/UAS) and

---

[2]Our code is publicly available at `https://github.com/DanielLeee/drtreesp`.
[3]`https://github.com/yzhangcs/parser`

Table 1: Comparison of mean UAS and execution time under different training set sizes. Time refers to the CPU time taken to finish one gradient descent step. Statistically significant differences compared to *BiAF* are marked with † (paired t-test, $p < 0.05$). The best UAS are highlighted in bold.

| Method | Time (s) | PTB | | | | CTB | | | | UD Dutch | | | | UD Turkish (low resource) | | | |
|---|---|---|---|---|---|---|---|---|---|---|---|---|---|---|---|---|---|
| | | m = 10 | 50 | 100 | 1000 | m = 10 | 50 | 100 | 1000 | m = 10 | 50 | 100 | 1000 | m = 10 | 50 | 100 | 1000 |
| BiAF | 0.34 | 93.48 | **96.87** | **96.95** | **97.16** | 88.45 | 90.89 | 91.15 | **91.70** | 90.86 | 93.80 | 94.15 | 94.98 | 17.64 | 26.59 | 30.75 | 42.82 |
| Marginal | 0.28 | 94.51† | 96.81† | 96.92 | 97.12 | 89.19† | 91.03† | 91.27 | 91.67 | **92.41**† | 94.22† | 94.50† | **95.15**† | 24.85† | **32.83**† | **33.75**† | **43.18** |
| Stochastic | 2.72 | **94.62**† | 96.81 | 96.93 | 97.14 | **89.27**† | 91.03† | **91.27** | 91.66 | 92.40† | 94.23† | 94.47 | 95.14† | **25.06**† | 31.35† | 33.62† | 41.20† |
| Game | 7.25 | 94.51† | 96.86 | 96.92 | 97.08† | 89.22† | **91.06**† | 91.22 | 91.57† | 92.32† | **94.34**† | **94.59**† | 95.01 | 19.85 | 23.18† | 27.12† | 36.30† |

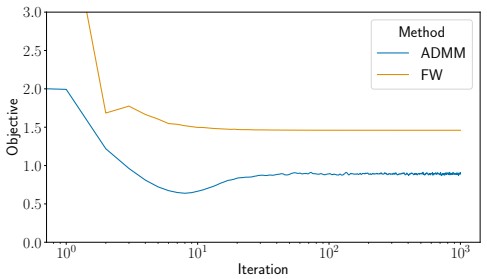

Figure 1: Convergence of ADMM and FW for random points with 95% confidence intervals.

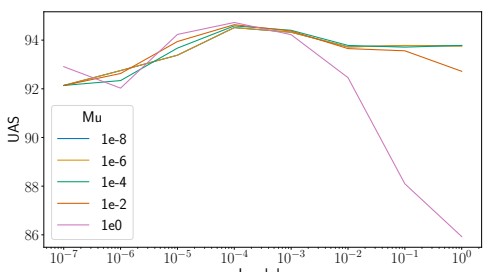

Figure 2: The best UAS with the Marginal algorithm as $\mu$ and $\lambda$ vary in logarithmic scales.

labeled/unlabeled complete matches (LCM/UCM). The attachment score can be transformed to the Hamming loss with linear mapping: $\text{AS}(\boldsymbol{y}, \boldsymbol{y}') \triangleq |\mathcal{V}| - 1 - \ell(\boldsymbol{y}, \boldsymbol{y}')/2$.

Full batch learning is adopted for *Marginal* (Eq. (3)). Mini-batch training is adopted for *Game*, the game-theoretic algorithm, and *Stochastic*, which solves the inner min-max problem in Eq. (2) using Eq. (3) with fixed $\boldsymbol{\theta}$. All models are trained with the training set only. The optimal hyperparameters and parameters are chosen based on the validation set. See Appendix C for detailed parameter values.

To showcase the ability of DRO methods tackling scarce data, in each run, we randomly draw $m \in \{10, 50, 100, 1000\}$ samples without replacement from the training set and keep the original validation and test sets. All the models are trained on the same set of sampled data. The process is repeated 5 times for each $m$. The main UAS results on the PTB, CTB and UD Dutch Lassy Small datasets are reported in Table 1 with complete results provided in Appendix C. Our methods consistently deliver higher UAS than *BiAF* especially with a small amount of data[4]. With little training data, DRO approaches minimize the worst-case risk to avoid overfitting. With more training data available, our method is still comparable to *BiAF* which is not significantly better than our methods by statistical tests. This illustrates the advantages of replacing conditional log-likelihood with our Fisher consistent surrogate loss without changing the number of model parameters. Moreover, we study a low-resource setting with the UD Turkish dataset in which only the sampled data is used for pretraining without BERT embeddings. The binary cross-entropy loss (single normalization) is adopted during pretraining in this setting to avoid pretrained features biased towards the multi-class cross-entropy loss (local normalization) adopted by *BiAF*. We observe consistently competitive performance of our methods in the low-resource setting in Table 1 as well.

We report computational time of one gradient descent step in the second column of Table 1, averaged across 10 runs. For fair comparisons, all the models are run with CPU only, with a batch size of 200. All the methods achieve their optimal validation set performance in 150-300 steps. *BiAF* and *Marginal* are the fastest because the most time-consuming step of computing dot products of features and parameters is only performed once whereas the other two methods perform it multiple times. However, since *Marginal* is unable to leverage stochastic gradients, its execution time grows linearly in the full batch size. Henceforth, there is a trade-off between *Marginal* and *Stochastic*/*Game* for computational efficiency. The extra cost compared to *BiAF* with cross entropy is expected because distributional robustness against a set of adversarial distributions is guaranteed.

---

[4]The UAS is high with 10 training samples possibly because (1) the backbone sub-network and linear layer were trained together with the whole training set; (2) BERT embeddings yield data representation that is easily linearly separable; (3) 10 samples result in as many as $10 \times 20 \times 20$ balanced head-selection instances for *BiAF*.

We compare ADMM and FW by performing for 100 times projection of random points in $[-5, 5]^{75}$ on a graph with 5 nodes and 3 parallel arcs between each $(i, j)$. We subtract the integral part of the observed minimum values in each run for better illustration. As shown in Figure 1, ADMM usually finds a better solution in the arborescence polytope than FW does within 1000 iterations[5]. That being said, the per-iteration cost of ADMM is about $8n$ times higher than that of FW due to consensus optimization of $n$ subproblems. In practice, the solution computed with FW usually leads to an approximately good sub-derivative to optimize the DRO objective. We have verified that the solutions suggested by ADMM satisfy the polytope constraints for graphs of up to 10 nodes.

We conduct sensitivity analysis by varying $\mu$ and $\lambda$ on UD Dutch with 100 training samples. Figure 2 implies that moderate smoothing is beneficial to generalization. The ambiguity radius should be judiciously chosen because a small $\lambda$ causes overfitting while a large $\lambda$ leads to conservative models.

## 7    Discussion and Conclusion

We proposed a distributionally robust and consistent tree structured prediction method. We showed its equivalence to regularized surrogate loss minimization. We put forward a provably convergent algorithm based on efficient projection oracles for arborescence polytopes. Our proposed method enjoys Fisher consistency and robustness against noise in conditional distributions in terms of feature moments. Theoretical and empirical results validate its effectiveness.

We assume that an expressive feature mapping is given such that a sufficiently good linear discriminant rule can be learned. The class-sensitive form $\phi(x, y)$ is general but consumes more memory than the decomposable form $\phi_X(x) \otimes \phi_Y(y)$. The ADMM projection algorithm is efficient theoretically with high per-iteration costs in practice. We expect this work to be a principled way of learning to predict tree-structured objects. Future directions include a more efficient implementation and general structured prediction with DRO. Potential negative societal impacts of our work include using its prediction without verification to guide human-centered design in policy-making.

**Representation learning.** Our method can be easily adapted to a representation learning framework with automatic differentiation. Although this may lead to a non-convex problem without the theoretical guarantees derived in this paper, it is highly desired in practice if feature mappings are optimized as well. We discuss a possible approach as follows. Modern neural networks for supervised learning typically have a linear layer in the end without activation. Assume the penultimate layer outputs $\Phi(x)$ for input $x$, the last layer with parameters $\theta$ will typically output $\psi(x) := \Phi(x)\theta \in \mathbb{R}^k$, sometimes called logits, with $k = n^2$ labels for all arcs when parsing a sentence of $n$ tokens. Note that $\theta$ in our formulation naturally serves as the parameters of this linear layer. Moreover, knowing $\psi(x)$ is sufficient for us to solve the inner minimax problem in Eq. (2) to get $\mathbb{P}^*_{\hat{Y}|x}$ and $\mathbb{Q}^*_{\hat{Y}|x}$. In this way, our DRO method can be considered a loss layer without learnable parameters, which backpropagates the sub-derivative of the objective with respect to $\psi(x)$:

$$\frac{\partial}{\partial \psi(x)} \ell_{\text{adv}} \in \frac{1}{B} \sum_{i=1}^{B} (q^{(i)*} - p^{(i)*}_{\text{emp}}),$$

where $B$ is the batch size. The sub-derivative of the regularization term with respect to $\theta$ should be added to the linear layer. Now we are able to take advantage of automatic differentiation and focus on solving the inner adversarial problem given $\psi(x)$ and $y$. Since the computational bottleneck lies in computing $\psi(x)$ and backward passes, the overhead of computing the adversarial loss may be dominated and not significant compared to the cross-entropy loss. We leave investigations on its effective applications to future work.

## Acknowledgments and Disclosure of Funding

This material is based upon work supported by the National Science Foundation under Grant Nos. 1652530, 1910146, and 1934915.

---

[5]One explanation is that FW relies on first-order approximations while there are exponential number of facets in the arborescence polytope.

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
