# A  Technical Proofs

**Proposition 1.** *The distributionally robust tree structured prediction problem based on moment divergence in Eq.* (1) *can be rewritten as*

$$\min_{\boldsymbol{\theta}} \mathbb{E}_{\mathbb{P}_{\boldsymbol{X},\boldsymbol{Y}}^{emp}} \underbrace{\min_{\mathbb{P}} \max_{\mathbb{Q}} \mathbb{E}_{\mathbb{P}_{\hat{\boldsymbol{Y}}|\boldsymbol{X}},\mathbb{Q}_{\check{\boldsymbol{Y}}|\boldsymbol{X}}} \ell(\hat{\boldsymbol{Y}},\check{\boldsymbol{Y}}) + \boldsymbol{\theta}^{\mathsf{T}}(\boldsymbol{\phi}(\boldsymbol{X},\check{\boldsymbol{Y}}) - \boldsymbol{\phi}(\boldsymbol{X},\boldsymbol{Y})) + \varepsilon\|\boldsymbol{\theta}\|_*}_{\ell_{adv}(\boldsymbol{\theta},(\boldsymbol{X},\boldsymbol{Y}))},$$

*where $\boldsymbol{\theta} \in \mathbb{R}^d$ is the vector of Lagrangian multipliers and $\|\cdot\|_*$ is the dual norm of $\|\cdot\|$.*

*Proof.*  Recall the primal problem

$$\min_{\mathbb{P}} \max_{\mathbb{Q}\in\mathcal{B}(\mathbb{P}^{emp})} \mathbb{E}_{\mathbb{Q}_{\boldsymbol{X},\check{\boldsymbol{Y}}}\mathbb{P}_{\hat{\boldsymbol{Y}}|\boldsymbol{X}}} \ell(\hat{\boldsymbol{Y}},\check{\boldsymbol{Y}}),$$

where $\mathcal{B}(\mathbb{P}^{\text{emp}}) := \{\mathbb{Q} : \mathbb{Q}_{\boldsymbol{X}} = \mathbb{P}_{\boldsymbol{X}}^{\text{emp}} \wedge \|\mathbb{E}_{\mathbb{P}^{\text{emp}}}\boldsymbol{\phi}(\cdot) - \mathbb{E}_{\mathbb{Q}}\boldsymbol{\phi}(\cdot)\| \leq \varepsilon\}$ with $\varepsilon \geq 0$.

Note the feature function $\boldsymbol{\phi}(\cdot)$ is fixed and given. Since $\mathbb{P}_{\hat{\boldsymbol{Y}}|\boldsymbol{X}} \in \Delta$ and $\mathbb{Q}_{\boldsymbol{X},\check{\boldsymbol{Y}}} \in \Delta \cap \mathcal{B}(\mathbb{P}^{\text{emp}})$ where $\Delta$ is the probability simplex with dimension omitted, the constraint sets are convex. The objective function is convex in $\mathbb{P}$ and concave in $\mathbb{Q}$ because it is affine in both. Therefore strong duality holds:

$$\max_{\mathbb{Q}\in\mathcal{B}(\mathbb{P}^{emp})} \min_{\mathbb{P}} \mathbb{E}_{\mathbb{Q}_{\boldsymbol{X},\check{\boldsymbol{Y}}}\mathbb{P}_{\hat{\boldsymbol{Y}}|\boldsymbol{X}}} \ell(\hat{\boldsymbol{Y}},\check{\boldsymbol{Y}}).$$

Let $\mathcal{C} := \{\boldsymbol{u} : \|\boldsymbol{u} - \mathbb{E}_{\mathbb{P}^{\text{emp}}}\boldsymbol{\phi}(\cdot)\| \leq \varepsilon\}$. Rewrite the problem with this constraint:

$$\sup_{\mathbb{Q},\boldsymbol{u}} \min_{\mathbb{P}} \mathbb{E}_{\mathbb{P}_{\boldsymbol{X}}^{emp}\mathbb{Q}_{\check{\boldsymbol{Y}}|\boldsymbol{X}}\mathbb{P}_{\hat{\boldsymbol{Y}}|\boldsymbol{X}}} \ell(\hat{\boldsymbol{Y}},\check{\boldsymbol{Y}}) - I_{\mathcal{C}}(\boldsymbol{u})$$

$$\text{s.t.} \quad \boldsymbol{u} = \mathbb{E}_{\mathbb{P}_{\boldsymbol{X}}^{emp}\mathbb{Q}_{\check{\boldsymbol{Y}}|\boldsymbol{X}}} \boldsymbol{\phi}(\boldsymbol{X},\check{\boldsymbol{Y}}),$$

where $I_{\mathcal{C}}(\cdot)$ is the indicator function with $I_{\mathcal{C}}(\boldsymbol{x}) = 0$ if $\boldsymbol{x} \in \mathcal{C}$ and $+\infty$ otherwise. The simplex constraints are omitted.

The dual problem by relaxing the equality constraint is

$$\sup_{\mathbb{Q},\boldsymbol{u}} \min_{\boldsymbol{\theta}} \min_{\mathbb{P}} \mathbb{E}_{\mathbb{P}_{\boldsymbol{X}}^{emp}\mathbb{Q}_{\check{\boldsymbol{Y}}|\boldsymbol{X}}\mathbb{P}_{\hat{\boldsymbol{Y}}|\boldsymbol{X}}} \ell(\hat{\boldsymbol{Y}},\check{\boldsymbol{Y}}) - I_{\mathcal{C}}(\boldsymbol{u}) + \boldsymbol{\theta}^{\mathsf{T}}\mathbb{E}_{\mathbb{P}_{\boldsymbol{X}}^{emp}\mathbb{Q}_{\check{\boldsymbol{Y}}|\boldsymbol{X}}}\boldsymbol{\phi}(\boldsymbol{X},\check{\boldsymbol{Y}}) - \boldsymbol{\theta}^{\mathsf{T}}\boldsymbol{u},$$

where $\boldsymbol{\theta}$ is the vector of Lagrange multipliers.

Given $\boldsymbol{X} = \boldsymbol{x}$, optimization of $\mathbb{Q}_{\check{\boldsymbol{Y}}|\boldsymbol{x}}$ and $\mathbb{P}_{\hat{\boldsymbol{Y}}|\boldsymbol{x}}$ can be done independently. Again by strong duality, we can rearrange the terms:

$$\min_{\boldsymbol{\theta}} \mathbb{E}_{\mathbb{P}_{\boldsymbol{X}}^{emp}} \min_{\mathbb{P}} \max_{\mathbb{Q}} \mathbb{E}_{\mathbb{Q}_{\check{\boldsymbol{Y}}|\boldsymbol{X}}\mathbb{P}_{\hat{\boldsymbol{Y}}|\boldsymbol{X}}} \ell(\hat{\boldsymbol{Y}},\check{\boldsymbol{Y}}) + \boldsymbol{\theta}^{\mathsf{T}}\boldsymbol{\phi}(\boldsymbol{X},\check{\boldsymbol{Y}}) + \sup_{\boldsymbol{u}} -I_{\mathcal{C}}(\boldsymbol{u}) - \boldsymbol{\theta}^{\mathsf{T}}\boldsymbol{u}.$$

The associated dual norm $\|\cdot\|_*$ of the norm $\|\cdot\|$ is defined as

$$\|\boldsymbol{z}\|_* := \sup\{\boldsymbol{z}^{\mathsf{T}}\boldsymbol{x} : \|\boldsymbol{x}\| \leq 1\},$$

based on which we are able to simplify the optimization over $\boldsymbol{u}$ as

$$\sup_{\boldsymbol{u}} -I_{\mathcal{C}}(\boldsymbol{u}) - \boldsymbol{\theta}^{\mathsf{T}}\boldsymbol{u} = \sup_{\boldsymbol{u}\in\mathcal{C}} -\boldsymbol{\theta}^{\mathsf{T}}\boldsymbol{u} = \sup_{\boldsymbol{e}:\|\boldsymbol{e}\|\leq 1} -\boldsymbol{\theta}^{\mathsf{T}}(\mathbb{E}_{\mathbb{P}^{emp}}\boldsymbol{\phi}(\cdot) - \varepsilon\boldsymbol{e}) = -\boldsymbol{\theta}^{\mathsf{T}}\mathbb{E}_{\mathbb{P}^{emp}}\boldsymbol{\phi}(\cdot) + \varepsilon\|\boldsymbol{\theta}\|_*.$$

Plugging it back to the dual problem, we have

$$\min_{\boldsymbol{\theta}} \mathbb{E}_{\mathbb{P}_{\boldsymbol{X},\boldsymbol{Y}}^{emp}} \min_{\mathbb{P}} \max_{\mathbb{Q}} \mathbb{E}_{\mathbb{Q}_{\check{\boldsymbol{Y}}|\boldsymbol{X}}\mathbb{P}_{\hat{\boldsymbol{Y}}|\boldsymbol{X}}} \ell(\hat{\boldsymbol{Y}},\check{\boldsymbol{Y}}) + \boldsymbol{\theta}^{\mathsf{T}}(\boldsymbol{\phi}(\boldsymbol{X},\check{\boldsymbol{Y}}) - \boldsymbol{\phi}(\boldsymbol{X},\boldsymbol{Y})) + \varepsilon\|\boldsymbol{\theta}\|_*.$$

$\square$

**Theorem 2.** *Given $m$ samples, a non-negative loss $\ell(\cdot,\cdot)$ such that $|\ell(\cdot,\cdot)| \leq K$, a feature function $\boldsymbol{\phi}(\cdot,\cdot)$ such that $\|\boldsymbol{\phi}(\cdot,\cdot)\| \leq B$, a positive ambiguity level $\varepsilon > 0$, then, for any $\rho \in (0,1]$, with a probability at least $1 - \rho$, the following excess true worst-case risk bound holds:*

$$\max_{\mathbb{Q}\in\mathcal{B}(\mathbb{P}^{true})} R_{\mathbb{Q}}^{L}(\boldsymbol{\theta}_{emp}^*) - \max_{\mathbb{Q}\in\mathcal{B}(\mathbb{P}^{true})} R_{\mathbb{Q}}^{L}(\boldsymbol{\theta}_{true}^*) \leq \frac{4KB}{\varepsilon\sqrt{m}}\left(1 + \frac{3}{2}\sqrt{\frac{\ln(4/\rho)}{2}}\right),$$

where $\boldsymbol{\theta}_{emp}^*$ and $\boldsymbol{\theta}_{true}^*$ are the optimal parameters learned in Eq. (2) under $\mathbb{P}^{emp}$ and $\mathbb{P}^{true}$ respectively. The original risk of $\boldsymbol{\theta}$ under $\mathbb{Q}$ is $R_{\mathbb{Q}}^L(\boldsymbol{\theta}) := \mathbb{E}_{\mathbb{Q}_{\boldsymbol{X},\boldsymbol{Y}}, \mathbb{P}_{\hat{\boldsymbol{Y}}|\boldsymbol{X}}^{\boldsymbol{\theta}}} \ell(\hat{\boldsymbol{Y}}, \boldsymbol{Y})$ with Bayes prediction $\mathbb{P}_{\boldsymbol{Y}|\boldsymbol{x}}^{\boldsymbol{\theta}} \in$ $\arg\min_{\mathbb{P}} \max_{\mathbb{Q}} \mathbb{E}_{\mathbb{Q}_{\check{\boldsymbol{Y}}|\boldsymbol{x}} \mathbb{P}_{\hat{\boldsymbol{Y}}|\boldsymbol{x}}} \ell(\hat{\boldsymbol{Y}}, \check{\boldsymbol{Y}}) + \boldsymbol{\theta}^{\mathsf{T}} \phi(\boldsymbol{x}, \check{\boldsymbol{Y}})$.

*Proof.* Define the adversarial surrogate risk of $\boldsymbol{\theta}$ with respect to $\tilde{\mathbb{P}}$ as

$$R_{\mathbb{P}}^S(\boldsymbol{\theta}) := \mathbb{E}_{\tilde{\mathbb{P}}_{\boldsymbol{X},\boldsymbol{Y}}} \ell_{\mathrm{adv}}(\boldsymbol{\theta},(\boldsymbol{X},\boldsymbol{Y})) := \mathbb{E}_{\tilde{\mathbb{P}}_{\boldsymbol{X},\boldsymbol{Y}}} \min_{\mathbb{P}} \max_{\mathbb{Q}} \mathbb{E}_{\mathbb{Q}_{\check{\boldsymbol{Y}}|\boldsymbol{X}} \mathbb{P}_{\hat{\boldsymbol{Y}}|\boldsymbol{X}}} \ell(\hat{\boldsymbol{Y}}, \check{\boldsymbol{Y}}) + \boldsymbol{\theta}^{\mathsf{T}}(\phi(\boldsymbol{X}, \check{\boldsymbol{Y}}) - \phi(\boldsymbol{X}, \boldsymbol{Y})) + \varepsilon\|\boldsymbol{\theta}\|_*.$$

Let $\boldsymbol{\theta}_{\mathrm{true}}^* \in \arg\min_{\boldsymbol{\theta}} R_{\mathbb{P}^{\mathrm{true}}}^S(\boldsymbol{\theta})$ and $\boldsymbol{\theta}_{\mathrm{emp}}^* \in \arg\min_{\boldsymbol{\theta}} R_{\mathbb{P}^{\mathrm{emp}}}^S(\boldsymbol{\theta})$ be the optimal parameters learned with $\mathbb{P}_{\boldsymbol{X},\boldsymbol{Y}}^{\mathrm{true}}$ and $\mathbb{P}_{\boldsymbol{X},\boldsymbol{Y}}^{\mathrm{emp}}$ respectively.

Given $\boldsymbol{x}$, define the decoded prediction by $\boldsymbol{\theta}$ as

$$\mathbb{P}_{\boldsymbol{Y}|\boldsymbol{x}}^{\boldsymbol{\theta}} \in \arg\min_{\mathbb{P}} \max_{\mathbb{Q}} \mathbb{E}_{\mathbb{Q}_{\check{\boldsymbol{Y}}|\boldsymbol{x}} \mathbb{P}_{\hat{\boldsymbol{Y}}|\boldsymbol{x}}} \ell(\hat{\boldsymbol{Y}}, \check{\boldsymbol{Y}}) + \boldsymbol{\theta}^{\mathsf{T}} \phi(\boldsymbol{x}, \check{\boldsymbol{Y}}).$$

Let the original risk of loss $\ell$ under some distribution $\mathbb{Q}$ be

$$R_{\mathbb{Q}}^L(\boldsymbol{\theta}) := \mathbb{E}_{\mathbb{Q}_{\boldsymbol{X},\boldsymbol{Y}}, \mathbb{P}_{\hat{\boldsymbol{Y}}|\boldsymbol{X}}^{\boldsymbol{\theta}}} \ell(\hat{\boldsymbol{Y}}, \boldsymbol{Y}).$$

According to Proposition 1, for any fixed $\mathbb{P}$, we have similarly

$$\max_{\mathbb{Q} \in \mathcal{B}(\mathbb{P}^{\mathrm{emp}})} \mathbb{E}_{\mathbb{Q}_{\boldsymbol{X},\check{\boldsymbol{Y}}} \mathbb{P}_{\hat{\boldsymbol{Y}}|\boldsymbol{X}}} \ell(\hat{\boldsymbol{Y}}, \check{\boldsymbol{Y}}) \triangleq \min_{\boldsymbol{\theta}} \mathbb{E}_{\mathbb{P}_{\boldsymbol{X},\boldsymbol{Y}}^{\mathrm{emp}}} \max_{\mathbb{Q}} \mathbb{E}_{\mathbb{Q}_{\check{\boldsymbol{Y}}|\boldsymbol{X}} \mathbb{P}_{\hat{\boldsymbol{Y}}|\boldsymbol{X}}} \ell(\hat{\boldsymbol{Y}}, \check{\boldsymbol{Y}}) + \boldsymbol{\theta}^{\mathsf{T}}(\phi(\boldsymbol{X}, \check{\boldsymbol{Y}}) - \phi(\boldsymbol{X}, \boldsymbol{Y})) + \varepsilon\|\boldsymbol{\theta}\|_*.$$

We start by looking at the worst-case risk of $\boldsymbol{\theta}_{\mathrm{true}}^*$ and $\boldsymbol{\theta}_{\mathrm{emp}}^*$.

$$\max_{\mathbb{Q} \in \mathcal{B}(\mathbb{P}^{\mathrm{true}})} R_{\mathbb{Q}}^L(\boldsymbol{\theta}_{\mathrm{emp}}^*)$$

$$= \min_{\boldsymbol{\theta}} \mathbb{E}_{\mathbb{P}_{\boldsymbol{X},\boldsymbol{Y}}^{\mathrm{true}}} \max_{\mathbb{Q}} \mathbb{E}_{\mathbb{Q}_{\check{\boldsymbol{Y}}|\boldsymbol{X}} \mathbb{P}_{\hat{\boldsymbol{Y}}|\boldsymbol{X}}^{\boldsymbol{\theta}_{\mathrm{emp}}^*}} \ell(\hat{\boldsymbol{Y}}, \check{\boldsymbol{Y}}) + \boldsymbol{\theta}^{\mathsf{T}}(\phi(\boldsymbol{X}, \check{\boldsymbol{Y}}) - \phi(\boldsymbol{X}, \boldsymbol{Y})) + \varepsilon\|\boldsymbol{\theta}\|_*$$

$$\leq \mathbb{E}_{\mathbb{P}_{\boldsymbol{X},\boldsymbol{Y}}^{\mathrm{true}}} \max_{\mathbb{Q}} \mathbb{E}_{\mathbb{Q}_{\check{\boldsymbol{Y}}|\boldsymbol{X}} \mathbb{P}_{\hat{\boldsymbol{Y}}|\boldsymbol{X}}^{\boldsymbol{\theta}_{\mathrm{emp}}^*}} \ell(\hat{\boldsymbol{Y}}, \check{\boldsymbol{Y}}) + \boldsymbol{\theta}_{\mathrm{emp}}^* \cdot (\phi(\boldsymbol{X}, \check{\boldsymbol{Y}}) - \phi(\boldsymbol{X}, \boldsymbol{Y})) + \varepsilon\|\boldsymbol{\theta}_{\mathrm{emp}}^*\|_*,$$

where the last inequality holds because $\boldsymbol{\theta}_{\mathrm{emp}}^*$ is not necessarily a minimizer. Similarly for $\boldsymbol{\theta}_{\mathrm{true}}^*$,

$$\max_{\mathbb{Q} \in \mathcal{B}(\mathbb{P}^{\mathrm{true}})} R_{\mathbb{Q}}^L(\boldsymbol{\theta}_{\mathrm{true}}^*) \leq \mathbb{E}_{\mathbb{P}_{\boldsymbol{X},\boldsymbol{Y}}^{\mathrm{true}}} \max_{\mathbb{Q}} \mathbb{E}_{\mathbb{Q}_{\check{\boldsymbol{Y}}|\boldsymbol{X}} \mathbb{P}_{\hat{\boldsymbol{Y}}|\boldsymbol{X}}^{\boldsymbol{\theta}_{\mathrm{true}}^*}} \ell(\hat{\boldsymbol{Y}}, \check{\boldsymbol{Y}}) + \boldsymbol{\theta}_{\mathrm{true}}^* \cdot (\phi(\boldsymbol{X}, \check{\boldsymbol{Y}}) - \phi(\boldsymbol{X}, \boldsymbol{Y})) + \varepsilon\|\boldsymbol{\theta}_{\mathrm{true}}^*\|_*.$$

On the other hand,

$$\mathbb{E}_{\mathbb{P}_{\boldsymbol{X},\boldsymbol{Y}}^{\mathrm{true}}} \max_{\mathbb{Q}} \mathbb{E}_{\mathbb{Q}_{\check{\boldsymbol{Y}}|\boldsymbol{X}} \mathbb{P}_{\hat{\boldsymbol{Y}}|\boldsymbol{X}}^{\boldsymbol{\theta}_{\mathrm{true}}^*}} \ell(\hat{\boldsymbol{Y}}, \check{\boldsymbol{Y}}) + \boldsymbol{\theta}_{\mathrm{true}}^* \cdot (\phi(\boldsymbol{X}, \check{\boldsymbol{Y}}) - \phi(\boldsymbol{X}, \boldsymbol{Y})) + \varepsilon\|\boldsymbol{\theta}_{\mathrm{true}}^*\|_*$$

$$= \min_{\boldsymbol{\theta}} \mathbb{E}_{\mathbb{P}_{\boldsymbol{X},\boldsymbol{Y}}^{\mathrm{true}}} \min_{\mathbb{P}} \max_{\mathbb{Q}} \mathbb{E}_{\mathbb{Q}_{\check{\boldsymbol{Y}}|\boldsymbol{X}} \mathbb{P}_{\hat{\boldsymbol{Y}}|\boldsymbol{X}}} \ell(\hat{\boldsymbol{Y}}, \check{\boldsymbol{Y}}) + \boldsymbol{\theta}^{\mathsf{T}}(\phi(\boldsymbol{X}, \check{\boldsymbol{Y}}) - \phi(\boldsymbol{X}, \boldsymbol{Y})) + \varepsilon\|\boldsymbol{\theta}\|_*$$

$$= \min_{\mathbb{P}} \min_{\boldsymbol{\theta}} \mathbb{E}_{\mathbb{P}_{\boldsymbol{X},\boldsymbol{Y}}^{\mathrm{true}}} \max_{\mathbb{Q}} \mathbb{E}_{\mathbb{Q}_{\check{\boldsymbol{Y}}|\boldsymbol{X}} \mathbb{P}_{\hat{\boldsymbol{Y}}|\boldsymbol{X}}} \ell(\hat{\boldsymbol{Y}}, \check{\boldsymbol{Y}}) + \boldsymbol{\theta}^{\mathsf{T}}(\phi(\boldsymbol{X}, \check{\boldsymbol{Y}}) - \phi(\boldsymbol{X}, \boldsymbol{Y})) + \varepsilon\|\boldsymbol{\theta}\|_*$$

$$\leq \min_{\boldsymbol{\theta}} \mathbb{E}_{\mathbb{P}_{\boldsymbol{X},\boldsymbol{Y}}^{\mathrm{true}}} \max_{\mathbb{Q}} \mathbb{E}_{\mathbb{Q}_{\check{\boldsymbol{Y}}|\boldsymbol{X}} \mathbb{P}_{\hat{\boldsymbol{Y}}|\boldsymbol{X}}^{\boldsymbol{\theta}_{\mathrm{true}}^*}} \ell(\hat{\boldsymbol{Y}}, \check{\boldsymbol{Y}}) + \boldsymbol{\theta}^{\mathsf{T}}(\phi(\boldsymbol{X}, \check{\boldsymbol{Y}}) - \phi(\boldsymbol{X}, \boldsymbol{Y})) + \varepsilon\|\boldsymbol{\theta}\|_*$$

$$= \max_{\mathbb{Q} \in \mathcal{B}(\mathbb{P}^{\mathrm{true}})} R_{\mathbb{Q}}^L(\boldsymbol{\theta}_{\mathrm{true}}^*),$$

where the first equality holds according to the definition of $\boldsymbol{\theta}_{\mathrm{true}}^*$. The above two inequalities imply the equality:

$$\max_{\mathbb{Q} \in \mathcal{B}(\mathbb{P}^{\mathrm{true}})} R_{\mathbb{Q}}^L(\boldsymbol{\theta}_{\mathrm{true}}^*) = \mathbb{E}_{\mathbb{P}_{\boldsymbol{X},\boldsymbol{Y}}^{\mathrm{true}}} \max_{\mathbb{Q}} \mathbb{E}_{\mathbb{Q}_{\check{\boldsymbol{Y}}|\boldsymbol{X}} \mathbb{P}_{\hat{\boldsymbol{Y}}|\boldsymbol{X}}^{\boldsymbol{\theta}_{\mathrm{true}}^*}} \ell(\hat{\boldsymbol{Y}}, \check{\boldsymbol{Y}}) + \boldsymbol{\theta}_{\mathrm{true}}^* \cdot (\phi(\boldsymbol{X}, \check{\boldsymbol{Y}}) - \phi(\boldsymbol{X}, \boldsymbol{Y})) + \varepsilon\|\boldsymbol{\theta}_{\mathrm{true}}^*\|_*.$$

Therefore,

$$\max_{\mathbb{Q} \in \mathcal{B}(\mathbb{P}^{\mathrm{true}})} R_{\mathbb{Q}}^L(\boldsymbol{\theta}_{\mathrm{emp}}^*) - \max_{\mathbb{Q} \in \mathcal{B}(\mathbb{P}^{\mathrm{true}})} R_{\mathbb{Q}}^L(\boldsymbol{\theta}_{\mathrm{true}}^*)$$

$$\leq \mathbb{E}_{\mathbb{P}_{\boldsymbol{X},\boldsymbol{Y}}^{\mathrm{true}}} \max_{\mathbb{Q}} \mathbb{E}_{\mathbb{Q}_{\check{\boldsymbol{Y}}|\boldsymbol{X}} \mathbb{P}_{\hat{\boldsymbol{Y}}|\boldsymbol{X}}^{\boldsymbol{\theta}_{\mathrm{emp}}^*}} \ell(\hat{\boldsymbol{Y}}, \check{\boldsymbol{Y}}) + \boldsymbol{\theta}_{\mathrm{emp}}^* \cdot (\phi(\boldsymbol{X}, \check{\boldsymbol{Y}}) - \phi(\boldsymbol{X}, \boldsymbol{Y})) + \varepsilon\|\boldsymbol{\theta}_{\mathrm{emp}}^*\|_*$$

$$- \left( \mathbb{E}_{\mathbb{P}_{\boldsymbol{X},\boldsymbol{Y}}^{\mathrm{true}}} \max_{\mathbb{Q}} \mathbb{E}_{\mathbb{Q}_{\check{\boldsymbol{Y}}|\boldsymbol{X}} \mathbb{P}_{\hat{\boldsymbol{Y}}|\boldsymbol{X}}^{\boldsymbol{\theta}_{\mathrm{true}}^*}} \ell(\hat{\boldsymbol{Y}}, \check{\boldsymbol{Y}}) + \boldsymbol{\theta}_{\mathrm{true}}^* \cdot (\phi(\boldsymbol{X}, \check{\boldsymbol{Y}}) - \phi(\boldsymbol{X}, \boldsymbol{Y})) + \varepsilon\|\boldsymbol{\theta}_{\mathrm{true}}^*\|_* \right). \quad (5)$$

The main idea is thus to use uniform convergence bounds. Firstly, by substituting $\mathbb{Q} = \mathbb{P}^{\text{true}}$, note that

$$\min_{\mathbb{P}} \max_{\mathbb{Q}} \mathbb{E}_{\mathbb{Q}_{\check{Y}|X} \mathbb{P}_{\hat{Y}|X}} \ell(\hat{Y}, \check{Y}) + \boldsymbol{\theta}^{\mathsf{T}}(\boldsymbol{\phi}(X, \check{Y}) - \boldsymbol{\phi}(X, Y)) \geq \min_{\mathbb{P}} \mathbb{E}_{\mathbb{P}^{\text{true}}_{Y|X} \mathbb{P}_{\hat{Y}|X}} \ell(\hat{Y}, Y) \geq 0.$$

We can get an upper bound of the norm of any optimal solution $\boldsymbol{\theta}^*_{\text{true}}$ or $\boldsymbol{\theta}^*_{\text{emp}}$ as follows:

$$0 + \varepsilon\|\boldsymbol{\theta}^*_{\text{true}}\|_* \leq R^S_{\mathbb{P}^{\text{true}}}(\boldsymbol{\theta}^*_{\text{true}}) \leq R^S_{\mathbb{P}^{\text{true}}}(\mathbf{0}) \leq \mathbb{E}_{\mathbb{P}^{\text{true}}_{X,Y}} \min_{\mathbb{P}} \max_{\mathbb{Q}} \mathbb{E}_{\mathbb{Q}_{\check{Y}|X} \mathbb{P}_{\hat{Y}|X}} \ell(\hat{Y}, \check{Y}) \leq K \implies \|\boldsymbol{\theta}^*_{\text{true}}\|_* \leq \frac{K}{\varepsilon}.$$

Let $\psi(X, Y) := \boldsymbol{\theta}^{\mathsf{T}}\boldsymbol{\phi}(X, Y)$ and $\boldsymbol{\psi}_x := (\psi(x, y))_{y \in \mathcal{Y}}$. Define

$$f(\boldsymbol{\theta}, \tilde{\mathbb{P}}) := \mathbb{E}_{\tilde{\mathbb{P}}_{X,Y}} \min_{\mathbb{P}} \max_{\mathbb{Q}} \mathbb{E}_{\mathbb{Q}_{\check{Y}|X} \mathbb{P}_{\hat{Y}|X}} \ell(\hat{Y}, \check{Y}) + \boldsymbol{\theta}^{\mathsf{T}}(\boldsymbol{\phi}(X, \check{Y}) - \boldsymbol{\phi}(X, Y))$$

$$\triangleq \mathbb{E}_{\tilde{\mathbb{P}}_{X,Y}} \max_{\mathbb{Q}} \mathbb{E}_{\mathbb{Q}_{\check{Y}|X} \mathbb{P}^{\boldsymbol{\theta}}_{\hat{Y}|X}} \ell(\hat{Y}, \check{Y}) + \boldsymbol{\theta}^{\mathsf{T}}(\boldsymbol{\phi}(X, \check{Y}) - \boldsymbol{\phi}(X, Y))$$

$$\triangleq \mathbb{E}_{\tilde{\mathbb{P}}_{X,Y}} \max_{\mathbb{Q}} \mathbb{E}_{\mathbb{Q}_{\check{Y}|X} \mathbb{P}^{\boldsymbol{\theta}}_{\hat{Y}|X}} \ell(\hat{Y}, \check{Y}) + \psi(X, \check{Y}) - \psi(X, Y)$$

$$\triangleq g(\boldsymbol{\psi}, \tilde{\mathbb{P}}).$$

Let $\boldsymbol{q}_x \in \Delta$ be the probability vector of $\mathbb{Q}_{\check{Y}|x}$ and $\boldsymbol{e}_y$ be the standard basis vector with $y$-th entry equal to 1. We have that for any $(x, y)$,

$$\frac{\partial}{\partial \boldsymbol{\psi}_x} g(\boldsymbol{\psi}, \delta_{(x,y)}) \subseteq \text{Conv}(\{\boldsymbol{q}_x - \boldsymbol{e}_y : \boldsymbol{q}_x \in \Delta\}) \implies \|\frac{\partial}{\partial \boldsymbol{\psi}_x} g(\boldsymbol{\psi}, \delta_{(x,y)})\|_1 \leq \max_{\boldsymbol{q}_x \in \Delta} \|\boldsymbol{q}_x - \boldsymbol{e}_y\|_1 \leq 2,$$

where $\delta_{(x,y)}$ is the Dirac point measure. $g(\cdot, \tilde{\mathbb{P}})$ is therefore 2-Lipschitz with respect to the $\ell_1$ norm. As per the assumption, $\|\boldsymbol{\phi}(\cdot, \cdot)\| \leq B$. This further implies that

$$f(\boldsymbol{\theta}_1, \delta_{(x_1, y_1)}) - f(\boldsymbol{\theta}_2, \delta_{(x_2, y_2)}) \leq \frac{4KB}{\varepsilon} \quad \forall \boldsymbol{\theta}_1, \boldsymbol{\theta}_2, x_1, x_2, y_1, y_2 \quad \text{s.t.} \quad \|\boldsymbol{\theta}_i\|_* \leq \frac{K}{\varepsilon} \quad \forall i = 1, 2.$$

We then follow the proof of Theorem 3 in Farnia and Tse [2016]. According to Theorem 26.12 in Shalev-Shwartz and Ben-David [2014], by uniform convergence, for any $\rho \in (0, 2]$, with a probability at least $1 - \frac{\rho}{2}$,

$$f(\boldsymbol{\theta}^*_{\text{emp}}, \mathbb{P}^{\text{true}}) - f(\boldsymbol{\theta}^*_{\text{emp}}, \mathbb{P}^{\text{emp}}) \leq \frac{4KB}{\varepsilon\sqrt{m}} \left(1 + \sqrt{\frac{\ln(4/\rho)}{2}}\right).$$

According to the definition of $\boldsymbol{\theta}^*_{\text{true}}$, the following inequality holds:

$$f(\boldsymbol{\theta}^*_{\text{emp}}, \mathbb{P}^{\text{emp}}) + \varepsilon\|\boldsymbol{\theta}^*_{\text{emp}}\|_* - f(\boldsymbol{\theta}^*_{\text{true}}, \mathbb{P}^{\text{emp}}) - \varepsilon\|\boldsymbol{\theta}^*_{\text{true}}\|_* \leq 0.$$

Since $\boldsymbol{\theta}^*_{\text{true}}$ do not depend on samples, according to the Hoeffding's inequality, with a probability $1 - \rho/2$,

$$f(\boldsymbol{\theta}^*_{\text{true}}, \mathbb{P}^{\text{emp}}) - f(\boldsymbol{\theta}^*_{\text{true}}, \mathbb{P}^{\text{true}}) \leq \frac{2KB}{\varepsilon\sqrt{m}} \sqrt{\frac{\ln(4/\rho)}{2}}.$$

Applying the union bound to the above three inequations, with a probability $1 - \rho$, we have

$$f(\boldsymbol{\theta}^*_{\text{emp}}, \mathbb{P}^{\text{true}}) + \varepsilon\|\boldsymbol{\theta}^*_{\text{emp}}\|_* - f(\boldsymbol{\theta}^*_{\text{true}}, \mathbb{P}^{\text{true}}) - \varepsilon\|\boldsymbol{\theta}^*_{\text{true}}\|_* \leq \frac{4KB}{\varepsilon\sqrt{m}} \left(1 + \frac{3}{2}\sqrt{\frac{\ln(4/\rho)}{2}}\right).$$

As stated by Inequation (5), we conclude with the following excess risk bound:

$$\max_{\mathbb{Q} \in \mathcal{B}(\mathbb{P}^{\text{true}})} R^L_{\mathbb{Q}}(\boldsymbol{\theta}^*_{\text{emp}}) - \max_{\mathbb{Q} \in \mathcal{B}(\mathbb{P}^{\text{true}})} R^L_{\mathbb{Q}}(\boldsymbol{\theta}^*_{\text{true}}) \leq \frac{4KB}{\varepsilon\sqrt{m}} \left(1 + \frac{3}{2}\sqrt{\frac{\ln(4/\rho)}{2}}\right).$$

$\square$

**Corollary 3.** *When $\varepsilon = 0$, $\ell_{adv}$ is Fisher consistent with respect to $\ell$. Namely,*

$$\mathbb{P}_{\hat{Y}|X}^{\theta_{true}^*} \in \arg\min_{\mathbb{P}_{\hat{Y}|X}} \mathbb{E}_{\mathbb{P}_{X,Y}^{true}, \mathbb{P}_{\hat{Y}|X}} \ell(\hat{Y}, Y),$$

*where $\theta_{true}^*$ is learned with $\ell_{adv}$ and $\mathbb{P}^{true}$ as in Theorem 2.*

*Proof.* Our formulation differs from Nowak-Vila et al. [2020] in the fact that we allow probabilistic prediction to be ground truth. By defining $y^*(\mu)$ as the gold standard probabilistic prediction and $\mathcal{Y}$ as the set of all possible probabilistic predictions in Proposition C.2 in Nowak-Vila et al. [2020], we have

$$\mathbb{P}_{\hat{Y}|x}^{\theta_{true}^*} \in \text{Conv}(\arg\min_{\mathbb{P}_{\hat{Y}|x}} \mathbb{E}_{\mathbb{P}_{Y|x}^{true}, \mathbb{P}_{\hat{Y}|x}} \ell(\hat{Y}, Y)).$$

Therefore,

$$\mathbb{P}_{\hat{Y}|x}^{\theta_{true}^*} \in \arg\min_{\mathbb{P}_{\hat{Y}|x}} \mathbb{E}_{\mathbb{P}_{Y|x}^{true}, \mathbb{P}_{\hat{Y}|x}} \ell(\hat{Y}, Y).$$

$\square$

**Proposition 4.** *Let $\mathcal{G}$ be a multi-graph. $\mathcal{A}_{marb} \triangleq \mathcal{A}_{arb}$.*

*Proof.* We follow the proof of Friesen [2019] for simple graphs. Recall the definition of $\mathcal{A}_{marb}$:

$$\mathcal{A}_{marb} := \{z^r : \exists z \geq 0$$

$$\sum_{a \in \delta^-(j)} z_a^k = \mathbb{1}(j \neq k) \, \forall k, j \in \mathcal{V} \wedge \tag{6}$$

$$\sum_{a \in \mathcal{E}_{ij}'} z_a^k = \sum_{a \in \mathcal{E}_{ij}} z_a^r \quad \forall k \neq r, i, j \in \mathcal{V}\}. \tag{7}$$

On one hand, given a legal $r$-arborescence with characteristic vector $z^r$, Eq. (6) and Eq. (7) hold by the definition of arborescences. The equality also holds for a convex combination of the characteristic vectors of $r$-arborescences.

On the other hand, given $z \in \mathcal{A}_{marb}$. Consider Edmond's definition of $r$-arborescence polytope based on rank constraints:

$$\sum_{a \in S} x_a \leq |S| - 1 \quad \forall S \subset \mathcal{V} \text{ with } S \neq \emptyset \tag{8}$$

$$\sum_{a \in \delta^-(j)} x_a = \mathbb{1}(j \neq r) \, \forall j \in \mathcal{V} \tag{9}$$

$$x \geq 0.$$

We have Eq. (6) directly implies Eq. (9). According to Eq. (7),

$$\sum_{a \in S} z_a^r = \sum_{a \in S} z_a^u \quad \forall S \subseteq \mathcal{V} \wedge u \in \mathcal{V}.$$

Therefore,

$$\sum_{a \in S} z_a^r = \sum_{a \in S} z_a^u \leq \sum_{j \in S} \sum_{a \in \delta^-(j)} z_a^u = |S| - 1 \quad \forall S \subseteq \mathcal{V} \wedge u \in S,$$

which is exactly Eq. (8). $\square$

**Proposition 5.** *Let $\mathcal{G}$ be a multi-graph. $\mathcal{A}_{mdep} \triangleq \mathcal{A}_{dep}$.*

*Proof.* Recall the definition of $\mathcal{A}_{mdep}$:

$$\mathcal{A}_{mdep} := \{z^r : z^r \in \mathcal{A}_{marb} \wedge$$

$$\sum_{a \in \delta^+(r)} z_a^r = 1\}. \tag{10}$$

---

**Algorithm 1** Double Oracle Game Solver

---

**Input:** Lagrange multipliers $\boldsymbol{\theta}$; feature function $\boldsymbol{\phi}(\cdot, \cdot)$; initial set of trees $\{\boldsymbol{y}_{\text{initial}}\}$
**Output:** A sparse Nash equilibrium $(\hat{\mathcal{T}}, \check{\mathcal{T}}, \mathbb{P}, \mathbb{Q})$
Initialize $\hat{\mathcal{T}} \leftarrow \check{\mathcal{T}} \leftarrow \{\boldsymbol{y}_{\text{initial}}\}$
**repeat**
$\quad (\mathbb{P}, \hat{v}_{\text{Nash}}) \leftarrow \text{SolveZeroSumGame}_{\hat{\mathcal{T}}}(\ell, \boldsymbol{\theta}^{\mathsf{T}}\boldsymbol{\phi}, \hat{\mathcal{T}}, \check{\mathcal{T}})$
$\quad (\check{\boldsymbol{y}}_{\text{BR}}, \check{v}_{\text{BR}}) \leftarrow \text{FindBestResponse}(\ell, \boldsymbol{\theta}^{\mathsf{T}}\boldsymbol{\phi}, \mathbb{P}, \hat{\mathcal{T}})$
$\quad$ **if** $\hat{v}_{\text{Nash}} \neq \check{v}_{\text{BR}}$ **then**
$\quad\quad \check{\mathcal{T}} \leftarrow \check{\mathcal{T}} \cup \{\check{\boldsymbol{y}}_{\text{BR}}\}$
$\quad$ **end if**
$\quad (\mathbb{Q}, \check{v}_{\text{Nash}}) \leftarrow \text{SolveZeroSumGame}_{\check{\mathcal{T}}}(\ell, \boldsymbol{\theta}^{\mathsf{T}}\boldsymbol{\phi}, \hat{\mathcal{T}}, \check{\mathcal{T}})$
$\quad (\hat{\boldsymbol{y}}_{\text{BR}}, \hat{v}_{\text{BR}}) \leftarrow \text{FindBestResponse}(\ell, \boldsymbol{\theta}^{\mathsf{T}}\boldsymbol{\phi}, \mathbb{Q}, \check{\mathcal{T}})$
$\quad$ **if** $\check{v}_{\text{Nash}} \neq \hat{v}_{\text{BR}}$ **then**
$\quad\quad \hat{\mathcal{T}} \leftarrow \hat{\mathcal{T}} \cup \{\hat{\boldsymbol{y}}_{\text{BR}}\}$
$\quad$ **end if**
**until** $\hat{v}_{\text{Nash}} = \check{v}_{\text{BR}} = \check{v}_{\text{Nash}} = \hat{v}_{\text{BR}}$
**return** $(\hat{\mathcal{T}}, \check{\mathcal{T}}, \mathbb{P}, \mathbb{Q})$

---

On one hand, given a legal dependency tree $\boldsymbol{z}^r \in \mathcal{A}_{\text{dep}}$, it satisfies Eq. (6) and Eq. (7) by Proposition 4. It also satisfies Eq. (10) by the definition of $\mathcal{A}_{\text{dep}}$.

On the other hand, given $\boldsymbol{z}^r \in \mathcal{A}_{\text{mdep}}$, firstly, $\boldsymbol{z}^r$ must be in $\mathcal{A}_{\text{arb}}$ by Proposition 4, which implies that we can write it as a convex combination of $k$ $r$-arborescences vectors: $\boldsymbol{z}^r \triangleq \alpha_1 \boldsymbol{t}^1 + \alpha_2 \boldsymbol{t}^2 + \cdots + \alpha_k \boldsymbol{t}^k$. All of them are legal $r$-arborescences, so $\sum_{a \in \delta^+(r)} t_a^i \geq 1$ for all $i \in [k]$. Now if $\sum_{a \in \delta^+(r)} t_a^i > 1$ for some $i$, we would have a contradiction, $\sum_{a \in \delta^+(r)} z_a^r > 1$. $\qquad\square$

## B  Algorithm Details

The pseudo-code of the constraint generation algorithm proposed in Section 3.2 is illustrated in Algorithm 1.

## C  More on Experiments

We adopt three public datasets, the English Penn Treebank (PTB v3.0) [Marcus et al., 1993], the Penn Chinese Treebank (CTB v5.1) [Xue et al., 2002], the Dutch Lassy Small Treebank and the Turkish Treebank in Universal Dependencies (UD v2.3) [Nivre et al., 2016]. We follow conventions in Chen and Manning [2014], Dyer et al. [2015] to prepare our data. We make standard train/validation/test splits. We use Stanford Dependencies (SD v3.3.0) [De Marneffe and Manning, 2008] to convert dependencies in PTB and CTB. The predicted POS tags with Stanford POS tagger [Toutanova et al., 2003] are adopted for PTB whereas gold POS tags are adopted for CTB and UD. Punctuation is excluded during evaluation[6].

The pretrained models are trained with the suggested hyperparameters in SuPar. The pretrained models achieve $97.25\%$, $91.91\%$ and $94.78\%$ UAS on PTB, CTB and UD Dutch respectively, where RoBERTa [Liu et al., 2019], ELECTRA [Cui et al., 2020] and XLM-RoBERTa [Conneau et al., 2019] are adopted as encoders. No BERT embeddings are adopted for the UD Turkish dataset.

For our ADMM algorithm, we adopt the adaptive scheme of varying penalty parameters ($\tau_{\text{incr}} = \tau_{\text{decr}} = 1.1$, $\mu = 1$) in Boyd et al. [2011] and the stopping criterion ($\epsilon_{\text{tol}} = 10^{-2}$) for consensus ADMM in Xu et al. [2017]. In FW, the learning rate is set to $\frac{2}{t+2}$. The smoothness weight $\mu$ and ambiguity radius $\lambda = 2\varepsilon$ are tuned using a logarithmic scale on $[10^{-7}, 1]$. The batch size for the game-theoretic algorithm is 10. The batch size for *Stochastic* is 200. The error tolerance in *Game* is set to $10^{-2}$. In stochastic gradient training, we use Adam with $lr = 10^{-2}$, $\beta_1 = 0.9$,

---

[6]A token is a punctuation if its gold POS tag is space, semi-colon, comma or period for English and PU for Chinese.

$\beta_2 = 0.999$, $\epsilon = 10^{-8}$. In our experiments, for efficiency, we again adopt the FW algorithm for the outer maximization in *Marginal*.

Complete main experimental results including all the metrics are shown in Table 2.

## D  Extension Details

For the dependency tree polytope, recall that the dual problem of projection onto $\mathcal{U}'_r := \{\boldsymbol{x} : \boldsymbol{x} \in \mathcal{U}_r \wedge \sum_{a \in \delta^+(r)} x_a = 1\}$ is

$$\max_{\boldsymbol{\alpha}, \beta} \sum_{a \in \mathcal{E}} h_a(\boldsymbol{\alpha}, \beta) - \sum_{j \neq r} \alpha_j - \beta \quad \text{s.t. } h_a(\boldsymbol{\alpha}, \beta) = \begin{cases} w_a^2 & \gamma_a > 2w_a, \\ w_a \gamma_a - \gamma_a^2/4 & \gamma_a \leq 2w_a, \end{cases}$$

where $\gamma_{(i,j,l)} := \alpha_j + \mathbb{1}(i = r)\beta$. Following Zhang et al. [2010] similarly, we sort $2w_{(i,j,l)}$ for each $j$ and compute the optimal $\alpha_j^*$ with $\beta = 0$. Let the sorted $w$'s be $(w_1^{(j)}, \ldots, w_n^{(j)})$ for each $j$. We blend create a set $\{w_x^{(j)} - \alpha_j^*\}$ for all $j$ and $x$. Let the sorted sequence be $-\infty = t_1 < t_2 < \cdots < t_{n_t} = \infty$. The derivative with respect to $\beta$ is piecewise-linear in each interval $[t_k, t_{k+1}]$. Since the objective is concave in $\beta$, we can iterate over all the intervals or find the optimal $\beta^*$ with binary search.

For higher-order tree local polytopes, the central problem is the projection onto

$$\mathcal{U}_s := \{\boldsymbol{x} \in \mathbb{R}_{\geq 0}^{|\mathcal{R}|} : x_s \leq x_a \quad \forall a \in s\}.$$

The only variables of interest are $x_a$ and $x_s$, given $x_s$, the optimal $x_a$ is simply $x_a^* = \max(w_a, x_s)$. We can sort $(w_a, w_s)_{a \in s}$ and enumerate the range $x_s$ takes over this set.

## E  Wong's Arborescence Polytope

We introduce another extended formulation of the arborescence polytope based on a multi-commodity flow representation [Wong, 1980, Martins, 2012, Friesen, 2019] as follows, which may be of independent interest:

$$\sum_{a \in \delta^-(j)} x_a = \mathbb{1}(j \neq r) \quad \forall j \in \mathcal{V} \tag{11}$$

$$\sum_{a \in \delta^-(j)} f_a^k - \sum_{a \in \delta^+(j)} f_a^k = \mathbb{1}(j = k) - \mathbb{1}(j = r) \quad \forall k \in \mathcal{V} \setminus \{r\}, j \in \mathcal{V} \tag{12}$$

$$0 \leq f_a^k \leq x_a \quad \forall a \in \mathcal{E}, k \in \mathcal{V} \setminus \{r\}. \tag{13}$$

Thus we have the arborescence polytope:

$$\mathcal{A}_{\text{mc}} = \{\boldsymbol{x} \in \mathbb{R}^{|\mathcal{E}|} | \exists \boldsymbol{f} : (\boldsymbol{x}, \boldsymbol{f}) \text{ satisfy equations } (11) - (13)\}.$$

According to Martins [2012], Friesen [2019], $\mathcal{A}_{\text{mc}} \triangleq \mathcal{A}_{\text{arb}}$ instead of an outer polytope of $\mathcal{A}_{\text{arb}}$.

We are interested in the following quadratic programming problem with linear inequality constraints:

$$\min_{\boldsymbol{x} \in \mathcal{A}_{\text{mc}}} \|\boldsymbol{x} - \boldsymbol{w}\|_2^2.$$

We can reformulate the problem as

$$\min_{\boldsymbol{x}, \boldsymbol{u}} g(\boldsymbol{x}, \boldsymbol{u}) := \frac{1}{2}\|\boldsymbol{x} - \boldsymbol{w}\|_2^2 + \frac{1}{2}\|\boldsymbol{u} - \boldsymbol{w}\|_2^2 + I_{\mathcal{X}}(\boldsymbol{x}) + I_{\mathcal{U}}(\boldsymbol{u})$$

$$\text{s.t.} \quad \boldsymbol{x} = \boldsymbol{u}$$

$$\mathcal{X} := \{\boldsymbol{x} : \sum_{a \in \delta^-(j)} x_a = \mathbb{1}(j \neq r) \forall j \in \mathcal{V} \wedge x_a \geq 0 \forall a \in \mathcal{E}\}$$

$$\mathcal{U} := \{\boldsymbol{u} : \exists \boldsymbol{f} \sum_{a \in \delta^-(j)} f_a^k - \sum_{a \in \delta^+(j)} f_a^k = \mathbb{1}(j = k) - \mathbb{1}(j = r) \quad \forall k \in \mathcal{V} \setminus \{r\}, j \in \mathcal{V}$$

$$0 \leq f_a^k \leq u_a \quad \forall k \in \mathcal{V} \setminus \{r\}, a \in \mathcal{E}\}.$$

Table 2: Comparison of mean UAS, LAS, UCM and LCM under different training set sizes. Statistically significant differences compared to BiAF are marked with † (paired t-test, $p < 0.05$). We highlight in bold the best results among the four methods.

| Dataset | # train | Metric | BiAF | Marginal | Stochastic | Game |
|---|---|---|---|---|---|---|
| PTB | 10 | UAS | $93.48 \pm 2.30$ | $94.51 \pm 1.71$† | $\mathbf{94.62 \pm 1.60}$† | $94.51 \pm 1.75$† |
| | | LAS | $92.02 \pm 2.26$ | $93.04 \pm 1.69$† | $\mathbf{93.14 \pm 1.58}$† | $93.04 \pm 1.73$† |
| | | UCM | $47.17 \pm 10.28$ | $52.30 \pm 8.71$† | $\mathbf{52.62 \pm 8.18}$† | $52.50 \pm 8.60$† |
| | | LCM | $39.73 \pm 7.96$ | $43.63 \pm 6.71$† | $\mathbf{43.97 \pm 6.39}$† | $43.86 \pm 6.58$† |
| | 50 | UAS | $\mathbf{96.87 \pm 0.06}$ | $96.81 \pm 0.05$† | $96.81 \pm 0.05$ | $96.86 \pm 0.05$ |
| | | LAS | $\mathbf{95.34 \pm 0.06}$ | $95.28 \pm 0.05$† | $95.28 \pm 0.05$ | $95.33 \pm 0.05$ |
| | | UCM | $67.65 \pm 0.81$ | $67.38 \pm 0.62$ | $67.18 \pm 0.79$ | $\mathbf{67.73 \pm 0.64}$ |
| | | LCM | $\mathbf{55.46 \pm 0.59}$ | $54.93 \pm 0.56$† | $54.79 \pm 0.59$† | $55.17 \pm 0.49$ |
| | 100 | UAS | $\mathbf{96.95 \pm 0.05}$ | $96.92 \pm 0.06$ | $96.93 \pm 0.05$ | $96.92 \pm 0.03$ |
| | | LAS | $\mathbf{95.42 \pm 0.05}$ | $95.40 \pm 0.06$ | $95.40 \pm 0.04$ | $95.39 \pm 0.02$ |
| | | UCM | $\mathbf{68.79 \pm 0.42}$ | $68.27 \pm 0.72$ | $68.36 \pm 0.41$ | $68.29 \pm 0.34$ |
| | | LCM | $\mathbf{56.21 \pm 0.14}$ | $55.68 \pm 0.56$ | $55.67 \pm 0.45$ | $55.66 \pm 0.33$ |
| | 1000 | UAS | $\mathbf{97.16 \pm 0.02}$ | $97.12 \pm 0.03$ | $97.14 \pm 0.02$ | $97.08 \pm 0.03$† |
| | | LAS | $\mathbf{95.63 \pm 0.03}$ | $95.59 \pm 0.02$ | $95.60 \pm 0.02$ | $95.55 \pm 0.03$† |
| | | UCM | $\mathbf{70.99 \pm 0.23}$ | $70.59 \pm 0.49$ | $70.61 \pm 0.32$ | $69.94 \pm 0.34$† |
| | | LCM | $\mathbf{57.57 \pm 0.09}$ | $57.18 \pm 0.28$† | $57.24 \pm 0.28$† | $56.80 \pm 0.23$† |
| CTB | 10 | UAS | $88.45 \pm 0.67$ | $89.19 \pm 0.38$† | $\mathbf{89.27 \pm 0.33}$† | $89.22 \pm 0.39$† |
| | | LAS | $84.79 \pm 0.62$ | $85.50 \pm 0.35$† | $\mathbf{85.58 \pm 0.30}$† | $85.53 \pm 0.36$† |
| | | UCM | $35.21 \pm 1.67$ | $36.83 \pm 1.20$ | $\mathbf{37.14 \pm 0.94}$† | $36.95 \pm 1.23$† |
| | | LCM | $25.86 \pm 0.87$ | $26.82 \pm 0.62$ | $\mathbf{26.95 \pm 0.59}$† | $26.95 \pm 0.63$† |
| | 50 | UAS | $90.89 \pm 0.10$ | $91.03 \pm 0.05$† | $91.03 \pm 0.05$† | $\mathbf{91.06 \pm 0.05}$† |
| | | LAS | $87.08 \pm 0.10$ | $87.20 \pm 0.05$† | $87.20 \pm 0.05$† | $\mathbf{87.23 \pm 0.06}$† |
| | | UCM | $42.54 \pm 0.24$ | $42.92 \pm 0.24$† | $42.86 \pm 0.12$† | $\mathbf{42.99 \pm 0.30}$ |
| | | LCM | $29.70 \pm 0.23$ | $29.69 \pm 0.36$ | $29.72 \pm 0.38$ | $\mathbf{29.79 \pm 0.23}$ |
| | 100 | UAS | $91.15 \pm 0.16$ | $91.27 \pm 0.08$ | $\mathbf{91.27 \pm 0.10}$ | $91.22 \pm 0.05$ |
| | | LAS | $87.32 \pm 0.14$ | $87.42 \pm 0.06$ | $\mathbf{87.42 \pm 0.08}$ | $87.37 \pm 0.05$ |
| | | UCM | $43.41 \pm 0.35$ | $\mathbf{43.91 \pm 0.27}$† | $43.86 \pm 0.43$† | $43.81 \pm 0.22$ |
| | | LCM | $30.02 \pm 0.22$ | $\mathbf{30.27 \pm 0.25}$ | $30.23 \pm 0.28$ | $30.26 \pm 0.26$ |
| | 1000 | UAS | $\mathbf{91.70 \pm 0.04}$ | $91.67 \pm 0.03$ | $91.66 \pm 0.03$ | $91.57 \pm 0.03$† |
| | | LAS | $\mathbf{87.84 \pm 0.04}$ | $87.80 \pm 0.03$ | $87.79 \pm 0.03$ | $87.70 \pm 0.03$† |
| | | UCM | $\mathbf{45.80 \pm 0.27}$ | $45.43 \pm 0.11$† | $45.41 \pm 0.12$† | $45.36 \pm 0.27$† |
| | | LCM | $31.14 \pm 0.19$ | $31.11 \pm 0.18$ | $31.08 \pm 0.17$ | $\mathbf{31.20 \pm 0.11}$ |
| UD Dutch | 10 | UAS | $90.86 \pm 1.23$ | $\mathbf{92.41 \pm 0.94}$† | $92.40 \pm 0.91$† | $92.32 \pm 1.03$† |
| | | LAS | $86.54 \pm 1.26$ | $\mathbf{88.10 \pm 0.95}$† | $88.08 \pm 0.91$† | $87.99 \pm 1.00$† |
| | | UCM | $64.11 \pm 2.18$ | $\mathbf{67.26 \pm 2.16}$† | $67.21 \pm 1.91$† | $67.26 \pm 1.97$† |
| | | LCM | $48.33 \pm 1.88$ | $50.32 \pm 1.75$† | $\mathbf{50.48 \pm 1.45}$† | $50.46 \pm 1.30$† |
| | 50 | UAS | $93.80 \pm 0.43$ | $94.22 \pm 0.26$† | $94.23 \pm 0.18$† | $\mathbf{94.34 \pm 0.24}$† |
| | | LAS | $89.36 \pm 0.33$ | $89.79 \pm 0.21$† | $89.79 \pm 0.12$† | $\mathbf{89.89 \pm 0.18}$† |
| | | UCM | $70.57 \pm 1.52$ | $72.42 \pm 0.90$† | $72.05 \pm 0.99$ | $\mathbf{72.60 \pm 1.39}$ |
| | | LCM | $52.40 \pm 0.61$ | $53.47 \pm 0.62$† | $53.40 \pm 0.59$ | $\mathbf{53.58 \pm 0.76}$ |
| | 100 | UAS | $94.15 \pm 0.18$ | $94.50 \pm 0.18$† | $94.47 \pm 0.13$ | $\mathbf{94.59 \pm 0.12}$† |
| | | LAS | $89.69 \pm 0.18$ | $90.04 \pm 0.15$† | $90.01 \pm 0.12$ | $\mathbf{90.12 \pm 0.10}$† |
| | | UCM | $71.71 \pm 0.92$ | $73.24 \pm 0.88$† | $73.01 \pm 0.99$ | $\mathbf{73.63 \pm 0.75}$† |
| | | LCM | $53.01 \pm 0.81$ | $53.79 \pm 0.40$ | $53.70 \pm 0.55$ | $\mathbf{54.13 \pm 0.44}$† |
| | 1000 | UAS | $94.98 \pm 0.07$ | $\mathbf{95.15 \pm 0.10}$† | $95.14 \pm 0.11$† | $95.01 \pm 0.05$ |
| | | LAS | $90.44 \pm 0.06$ | $\mathbf{90.59 \pm 0.08}$† | $90.59 \pm 0.08$† | $90.44 \pm 0.06$ |
| | | UCM | $74.73 \pm 0.33$ | $\mathbf{75.87 \pm 0.63}$† | $75.64 \pm 0.57$† | $75.41 \pm 0.56$ |
| | | LCM | $54.59 \pm 0.13$ | $\mathbf{55.21 \pm 0.17}$† | $55.16 \pm 0.21$† | $54.70 \pm 0.22$ |
| UD Turkish | 10 | UAS | $17.64 \pm 2.45$ | $24.85 \pm 2.35$† | $\mathbf{25.06 \pm 0.58}$† | $19.85 \pm 0.46$ |
| | | LAS | $4.86 \pm 2.74$ | $5.33 \pm 2.97$ | $\mathbf{5.40 \pm 2.85}$ | $5.02 \pm 3.04$ |
| | | UCM | $7.69 \pm 1.72$ | $9.03 \pm 1.33$ | $7.88 \pm 2.27$ | $\mathbf{10.03 \pm 0.54}$ |
| | | LCM | $1.46 \pm 1.03$ | $1.50 \pm 1.07$ | $1.50 \pm 1.07$ | $\mathbf{1.74 \pm 1.38}$ |
| | 50 | UAS | $26.59 \pm 2.37$ | $\mathbf{32.83 \pm 1.50}$† | $31.35 \pm 1.10$† | $23.18 \pm 2.03$† |
| | | LAS | $10.14 \pm 0.57$ | $10.73 \pm 0.86$ | $\mathbf{10.74 \pm 0.54}$ | $10.10 \pm 0.69$ |
| | | UCM | $10.03 \pm 1.31$ | $10.63 \pm 0.50$ | $\mathbf{10.81 \pm 0.50}$ | $10.34 \pm 0.36$ |
| | | LCM | $3.24 \pm 0.31$ | $3.26 \pm 0.24$ | $3.38 \pm 0.27$ | $\mathbf{3.43 \pm 0.27}$ |
| | 100 | UAS | $30.75 \pm 1.13$ | $\mathbf{33.75 \pm 0.86}$† | $33.62 \pm 1.49$† | $27.12 \pm 1.25$† |
| | | LAS | $10.84 \pm 0.80$ | $11.48 \pm 0.75$ | $\mathbf{11.69 \pm 0.67}$† | $10.48 \pm 0.70$† |
| | | UCM | $\mathbf{11.61 \pm 1.22}$ | $11.30 \pm 0.29$ | $11.34 \pm 0.26$ | $11.08 \pm 0.44$ |
| | | LCM | $3.53 \pm 0.60$ | $\mathbf{3.61 \pm 0.31}$ | $3.57 \pm 0.23$ | $3.55 \pm 0.23$ |
| | 1000 | UAS | $42.82 \pm 1.82$ | $\mathbf{43.18 \pm 1.73}$ | $41.20 \pm 2.17$† | $36.30 \pm 2.79$† |
| | | LAS | $\mathbf{18.44 \pm 1.00}$ | $18.24 \pm 1.62$ | $18.13 \pm 1.13$ | $16.38 \pm 1.20$† |
| | | UCM | $\mathbf{15.86 \pm 0.40}$ | $15.18 \pm 0.81$ | $13.78 \pm 0.30$† | $13.52 \pm 0.43$† |
| | | LCM | $\mathbf{4.49 \pm 0.47}$ | $4.37 \pm 0.46$ | $4.31 \pm 0.41$† | $4.29 \pm 0.38$† |

The scaled augmented Lagrangian function is

$$L_\rho(\boldsymbol{x}, \boldsymbol{u}, \boldsymbol{y}) = g(\boldsymbol{x}, \boldsymbol{u}) + \boldsymbol{\lambda}'^\mathsf{T}(\boldsymbol{x} - \boldsymbol{u}) + \frac{\rho}{2}\|\boldsymbol{x} - \boldsymbol{u}\|_2^2$$

$$= g(\boldsymbol{x}, \boldsymbol{u}) + \frac{\rho}{2}\|\boldsymbol{x} - \boldsymbol{u} + \frac{1}{\rho}\boldsymbol{\lambda}'\|_2^2 - \frac{1}{2\rho}\|\boldsymbol{\lambda}'\|_2^2$$

$$= g(\boldsymbol{x}, \boldsymbol{u}) + \frac{\rho}{2}\|\boldsymbol{x} - \boldsymbol{u} + \boldsymbol{\lambda}\|_2^2 - \frac{\rho}{2}\|\boldsymbol{\lambda}\|_2^2,$$

where $\qquad \boldsymbol{\lambda} := \frac{1}{\rho}\boldsymbol{\lambda}'.$

The ADMM algorithm updates the parameters as follows:

$$\boldsymbol{x}^{t+1} := \arg\min_{\boldsymbol{x}} L_\rho(\boldsymbol{x}, \boldsymbol{u}^t, \boldsymbol{\lambda}^t)$$

$$= \arg\min_{\boldsymbol{x}} \frac{1}{2}\|\boldsymbol{x} - \boldsymbol{w}\|_2^2 + I_{\mathcal{X}}(\boldsymbol{x}) + \frac{\rho}{2}\|\boldsymbol{x} - \boldsymbol{u}^t + \boldsymbol{\lambda}^t\|_2^2$$

$$= \arg\min_{\boldsymbol{x} \in \mathcal{X}} \|\boldsymbol{x} - \frac{1}{\rho+1}(\boldsymbol{w} + \rho\boldsymbol{u}^t - \rho\boldsymbol{\lambda}^t)\|_2^2,$$

$$\triangleq \operatorname{Proj}_{\mathcal{X}}(\frac{1}{\rho+1}(\boldsymbol{w} + \rho\boldsymbol{u}^t - \rho\boldsymbol{\lambda}^t))$$

$$\boldsymbol{u}^{t+1} := \arg\min_{\boldsymbol{u}} L_\rho(\boldsymbol{x}^{t+1}, \boldsymbol{u}, \boldsymbol{\lambda}^t)$$

$$= \arg\min_{\boldsymbol{u}} \frac{1}{2}\|\boldsymbol{u} - \boldsymbol{w}\|_2^2 + I_{\mathcal{F}}(\boldsymbol{u}) + \frac{\rho}{2}\|\boldsymbol{x}^{t+1} - \boldsymbol{u} + \boldsymbol{\lambda}^t\|_2^2$$

$$= \arg\min_{\boldsymbol{u} \in \mathcal{U}} \|\boldsymbol{u} - \frac{1}{\rho+1}(\boldsymbol{w} + \rho\boldsymbol{x}^{t+1} + \rho\boldsymbol{\lambda}^t)\|_2^2,$$

$$\triangleq \operatorname{Proj}_{\mathcal{U}}(\frac{1}{\rho+1}(\boldsymbol{w} + \rho\boldsymbol{x}^{t+1} + \rho\boldsymbol{\lambda}^t))$$

$$\boldsymbol{\lambda}^{t+1} := \boldsymbol{\lambda}^t + (\boldsymbol{x}^{t+1} - \boldsymbol{u}^{t+1}).$$

Projection onto $\mathcal{X}$ is decomposable over each $j \in \mathcal{V}$. And for each $j$, the optimal value of the group can be computed in $\mathcal{O}(n)$ in almost closed form via Section 5.5.1 in Zhang et al. [2010] or other simplex projection algorithms in $\mathcal{O}(n \log n)$.

Projection onto $\mathcal{U}$ is a minimum quadratic capacity expansion cost problem for fixed multi-commodity flows:

$$\min_{\boldsymbol{u} \in \mathcal{U}}\|\boldsymbol{u} - \boldsymbol{w}\|_2^2.$$

A partially relaxed problem is

$$\max_{\boldsymbol{\beta}} \min_{\boldsymbol{u}, \boldsymbol{f}}\|\boldsymbol{u} - \boldsymbol{w}\|_2^2 + \sum_{a,k} \beta_a^k(f_a^k - u_a)$$

$$\text{s.t.} \quad \sum_{a \in \delta^-(j)} f_a^k - \sum_{a \in \delta^+(j)} f_a^k = \mathbb{I}(j = k) - \mathbb{I}(j = r) \quad \forall k \in \mathcal{V} \setminus \{r\}, j \in \mathcal{V}$$

$$f_a^k \geq 0, \beta_a^k \geq 0 \quad \forall k \in \mathcal{V} \setminus \{r\}, a \in \mathcal{E}.$$

Given $\boldsymbol{\beta}$, the sub-problem for $\boldsymbol{u}$ is

$$\min_{\boldsymbol{u}} \sum_a u_a^2 - 2u_a w_a - \sum_k \beta_a^k u_a,$$

with an analytical solution

$$\boldsymbol{u}^* = \boldsymbol{w} + \frac{1}{2}\boldsymbol{\beta}^k.$$

Given $\boldsymbol{\beta}$, the sub-problem for $\boldsymbol{f}$ is

$$\min_{\boldsymbol{f}} \sum_{a,k} \beta_a^k f_a^k$$

$$\text{s.t.} \quad \sum_{a \in \delta^-(j)} f_a^k - \sum_{a \in \delta^+(j)} f_a^k = \mathbb{I}(j = k) - \mathbb{I}(j = r) \forall k \in \mathcal{V} \setminus \{r\}, j \in \mathcal{V}$$

$$f_a^k \geq 0 \quad \forall k \in \mathcal{V} \setminus \{r\}, a \in \mathcal{E},$$

which is a minimum-cost multi-commodity flow problem.

With $\boldsymbol{u}^*$ and $\boldsymbol{f}^*$, we can optimize $\boldsymbol{\beta}$ with sub-gradient ascent.

Alternatively, another partially relaxed problem is

$$\max_{\boldsymbol{\beta}} \min_{\boldsymbol{u},\boldsymbol{f}} \|\boldsymbol{u} - \boldsymbol{w}\|_2^2 + \sum_{a,k} f_a^k(\beta_{h(a)}^k - \beta_{t(a)}^k) + \sum_k \beta_r^k - \beta_k^k$$

$$\text{s.t.} \quad 0 \leq f_a^k \leq u_a, \beta_a^k \geq 0 \quad \forall k \in \mathcal{V} \setminus \{r\}, a \in \mathcal{E},$$

where $h(a)$ and $t(a)$ are the head and tail of arc $a$ respectively.

Given $\boldsymbol{\beta}$, the inner minimization problem is decomposed over $a$:

$$\min_{\boldsymbol{u},\boldsymbol{f}} u_a^2 - 2u_a w_a + \sum_k f_a^k(\beta_{h(a)}^k - \beta_{t(a)}^k)$$

$$\text{s.t.} \quad 0 \leq f_a^k \leq u_a \quad \forall k \in \mathcal{V} \setminus \{r\},$$

which is a convex continuous knapsack problem for each $a$.

The above optimization requires sub-gradient methods, which are usually slower than FW ($\mathcal{O}(\frac{1}{\epsilon^2})$).