# OpenReview forum: "Moment Distributionally Robust Tree Structured Prediction"
_NeurIPS.cc/2022/Conference — NeurIPS 2022 Accept_

### Official Review · Reviewer_ULtE · 2022-07-11

**Rating:** 7
**Confidence:** 4
**Soundness:** 4 excellent
**Presentation:** 4 excellent
**Contribution:** 3 good

**Summary:**

This work proposes a moment-based distributional robust optimization approach for tree structured prediction, which is further applied to dependency parsing. The authors propose the instantiation of the distributional robustness objective to the tree structured prediction setting in the parameterization of marginal probabilities. The authors also discuss the risk bound for the proposed methods and methods for projection to the arborescence polytopes. Experiments on three datasets demonstrate the effectiveness of the proposed method in the few-shot setting.

**Questions:**

See above discussions

**Limitations:**

See above discussions

**Strengths And Weaknesses:**

Strengths

- I like the rigorous mathematical discussions on this work. The authors give thorough discussions about the background and clear notes on the related work of multiple threads, and how this work is built on top on the existing methods.
- The idea of using distributionally robust optimization for tree structured prediction is indeed significant and would add valid contribution of the field
- Experiments show its effectiveness especially on low-resource settings

Weakness

- Ideally, distributionally robust optimization should be validated on datasets with distributional shifts (e.g., from news domain to dialog domain). But the evaluation is more on a in-distribution few-shot setting (rather than a distributional shift setting). I would expect the authors to elaborate this a little bit in the discussion period.
- The implementation is on CPUs with C and Python. It remains unclear how this method can be implemented on GPU-accelerated Automatic Differentiation libraries like PyTorch. Such limitation could severely restrict the applicability of the proposed method
- The gradients w.r.t. the projection step to the Polytope (section 4 and Figure 1) seems to be not well-defined and have previously been under discussions[1, 2]. It is not very clear in this paper’s setting how such problem may (or may not) become an issue in terms of optimization

[1] Peng et. al. ACL 2018. Backpropagating through Structured Argmax using a SPIGOT

[2] Mihaylova et. al. 2020. Understanding the Mechanics of SPIGOT: Surrogate Gradients for Latent Structure Learning

---

> ### Author Response · Authors · 2022-08-02
> **Response to Reviewer ULtE**
>
> Thanks for the positive comment.
>
> **Q: Ideally, distributionally robust optimization should be validated on datasets with distributional shifts (e.g., from news domain to dialog domain). But the evaluation is more on a in-distribution few-shot setting (rather than a distributional shift setting). I would expect the authors to elaborate this a little bit in the discussion period.**
>
> A: Good point. Generally speaking, distributionally robust optimization models the uncertainty about the true distribution with an ambiguity set. DRO methods may be sensitive to outliers if applied naively [4]. Existing DRO methods explicitly aimed for distributional shift settings usually make assumptions about the type of shifting and define a specific ambiguity set to deal with it. See [1, 2, 3, 4] for example. We do not make such assumptions but develop our method based on some similar formulations that have been shown to be Fisher consistent [5]. Specifically, we restrict the adversarial distribution to have the same marginal covariate distribution as the training distribution so there is no covariate shift robustness guarantee. However, the feature moment inequality ensures robustness against noises in conditional label distributions. The few-shot setting with few samples also makes the true distribution highly uncertain, which should resemble a distributional shift setting in some sense. Besides, Fisher consistency brought by the moment-based DRO formulation considered in this paper is one of our motivations and is generally absent in other kinds of DRO formulations in the literature.
>
> **Q: The implementation is on CPUs with C and Python. It remains unclear how this method can be implemented on GPU-accelerated Automatic Differentiation libraries like PyTorch. Such limitation could severely restrict the applicability of the proposed method.**
>
> A: In fact, we did adopt PyTorch to compute dot products between features and parameters by converting NumPy array data to PyTorch tensor data. Furthermore, our method can be easily adapted to an end-to-end automatic differentiation framework. Please see our next comment for details.
>
> **Q: The gradients w.r.t. the projection step to the Polytope (section 4 and Figure 1) seems to be not well-defined and have previously been under discussions[1, 2]. It is not very clear in this paper’s setting how such problem may (or may not) become an issue in terms of optimization.**
>
> A: The arborescence projection problem we study in Section 4 is well-defined because it is a convex optimization problem with a strongly convex objective function and convex constraints, implying a unique optimal solution. It is different from the referenced problems because our constraint is always convex and we do not elicit gradients with respect to the input of the projection problem. Instead, the projection is expected to be solved approximately well for faster convergence of the overall saddle-point problem. Even in backpropagation when our method is equipped with representation learning, the projection step is quite independent and only helps compute $\boldsymbol{q}^*$ to be used for real backpropagation. Figure 1 shows that ADMM generally finds a better solution than FW does, which we have verified to be valid convex combinations of arborescences for $n \leq 10$. One explanation is that FW relies on first-order approximations while there are an exponential number of facets in the polytope. Notwithstanding the sub-optimality, the solution computed with FW usually leads to an approximately good sub-derivative for the overall optimization problem in practice.
>
>
> [1] Zhang, Jingzhao, Aditya Krishna Menon, Andreas Veit, Srinadh Bhojanapalli, Sanjiv Kumar, and Suvrit Sra. "Coping with Label Shift via Distributionally Robust Optimisation." In International Conference on Learning Representations. 2020.
>
> [2] Sutter, Tobias, Andreas Krause, and Daniel Kuhn. "Robust Generalization despite Distribution Shift via Minimum Discriminating Information." Advances in Neural Information Processing Systems 34 (2021): 29754-29767.
>
> [3] Duchi, John C., Tatsunori Hashimoto, and Hongseok Namkoong. "Distributionally robust losses against mixture covariate shifts." Under review 2 (2019).
>
> [4] Zhai, Runtian, Chen Dan, Zico Kolter, and Pradeep Ravikumar. "Doro: Distributional and outlier robust optimization." In International Conference on Machine Learning, pp. 12345-12355. PMLR, 2021.
>
> [5] Fathony, Rizal, Sima Behpour, Xinhua Zhang, and Brian Ziebart. "Efficient and consistent adversarial bipartite matching." In International Conference on Machine Learning, pp. 1457-1466. PMLR, 2018.

---

> > ### Comment · Reviewer_ULtE · 2022-08-09
> > **Thank you for your clarification**
> >
> > I have improved my score according. I believe this paper will make a solid contribution in structured prediction.

---

> ### Author Response · Authors · 2022-08-02
> **Response to Reviewer ULtE (representation learning details)**
>
> **Representation Learning**. Incorporating automatic representation learning into our method is indeed highly desired because of its practical value in applications. We omitted the discussion of this topic in our initial submission due to space limits, but will include it in our revision if extra space is available.
> Although any representation learning model can be adopted, we focus on discussing the most popular one nowadays, the neural network model with end-to-end learning and automatic differentiation. We show how to make use of our DRO method as the final loss layer in a neural network model. A network for supervised learning typically has a linear classification layer in the end without activation. Assume the penultimate layer outputs $\boldsymbol{\Phi}(\boldsymbol{x}) \in \mathbb{R}^{k \times d}$ for input $\boldsymbol{x}$, the last layer will typically output  $\boldsymbol{\psi}(\boldsymbol{x}) := \boldsymbol{\Phi}(\boldsymbol{x}) \boldsymbol{\theta} \in \mathbb{R}^{k}$ for some $\boldsymbol{\theta} \in \mathbb{R}^{d}$. $\boldsymbol{\psi}(\boldsymbol{x})$ is sometimes called logits and yields probability distribution with a softmax layer. For example, in univariate classification, $k$ is the number of labels. In dependency parsing, $k = n^2$ with $n$ being the number of tokens in the input sentence $\boldsymbol{x}$. Given $b$-dimensional token-wise embeddings before the penultimate layer, the biaffine layer in BiAF yields $\boldsymbol{\Phi}(\boldsymbol{x}) \in \mathbb{R}^{n^2 \times b^2}$ equivalently ($b^2$-dimensional feature vector for each arc). Thus $\boldsymbol{\psi}(\boldsymbol{x}) \in \mathbb{R}^{n^2}$ is the logits for all the arcs. Note that $\boldsymbol{\theta}$ in our formulation is naturally equivalent to the parameters of the aforementioned last linear layer. Therefore having $\boldsymbol{\psi}(\boldsymbol{x})$ is sufficient for us to compute $\mathbb{P}^*_{Y|\boldsymbol{x}}$ and $\mathbb{Q}^*_{Y|\boldsymbol{x}}$. In this way, our method is the loss layer without learnable parameters, which backpropagates the sub-derivative of the objective with respect to $\boldsymbol{\psi}(\boldsymbol{x})$ to the linear classification layer: $\partial{\text{Obj}}/\partial{\boldsymbol{\psi}(\boldsymbol{x})} \triangleq \sum_{i = 1}^{B}  {\mathbf q}^{(i)*} - {\mathbf p}_{\text{emp}}^{(i)}$, where $B$ is the batch size. Recall $\mathbf{q}$ and ${\mathbf p}_\text{emp}$ are the probability vectors for $\mathbb{Q}$ and $\mathbb{P}^{\text{emp}}$ respectively. The sub-derivative of the regularization term with respect to $\boldsymbol{\theta}$ is added to the classification layer. Although losing global convergent and provable generalization guarantees, we are now able to take advantage of automatic differentiation and focus on solving the minimax problem given $\boldsymbol{\psi}(\boldsymbol{x})$ and groundtruth $\boldsymbol{y}$ for training. Since the computational bottleneck lies in computing $\boldsymbol{\Phi}(\boldsymbol{x}) \boldsymbol{\theta}$ while GPU acceleration now does it for us, the overhead of computing the adversarial loss should not be much higher than that of computing the cross-entropy loss.

---

### Official Review · Reviewer_sgAv · 2022-07-11

**Rating:** 7
**Confidence:** 3
**Soundness:** 3 good
**Presentation:** 3 good
**Contribution:** 3 good

**Summary:**

This paper attempts to develop a better learning algorithm for structured prediction tasks (in particular trees), the most direct application being supervised dependency parsing. Dependency parsing has a long line of work most of which are trained with maximum likelihood, margin-based, or ERM-based objectives. The authors point out that such training objectives are not aligned with the evaluation objective. Further, such methods have non-convex objectives due to which they lose convergence and generalization guarantees.

To mitigate that, the paper proposes a training approach based on the Distributionally Robust Optimization framework (DRO). In very simple terms, in DRO, the learning algorithm is unaware of the underlying distribution apart from some knowledge over the support and the goal is to minimize the worst-case risk over some uncertainty set. They show the equivalence between the DRO-based objective to a convex surrogate loss function and derive uniform convergence based generalization bounds. They develop efficient training algorithms and show convergence guarantees.

They evaluate their method on three dependency parsing datasets (PTB, UD, and CTB) with BiAF as the baseline. In particular, they focus on low training data settings (10 - 1000 training examples). They use their method to predict the unlabelled dependency tree and use a different classifier to predict arc labels. The model uses representation learned by Roberta and the final linear predictor is learned by their approach (or BiAF as baseline). Their approach achieves consistent gains over BiAF on the three datasets.


**Questions:**

(Q1) **Low resources languages.** Wouldn't it be more useful to evaluate/test the models on dependency parsing datasets for low resource languages? I understand that the training data here is restricted to 1000 examples but the pretrained model representations of high-resource languages seem to be good enough for BiAF to achieve over 85% with 10 examples. Wouldn't it be more useful to experiment on datasets that are not saturated (mainly low resource ones)?

(Q2) **Representation learning.** Is it feasible to design an algorithm with the proposed objective that also updates the feature representation (using some different optimization method)? In that, I guess convergence guarantees and generalization bounds may not hold but may lead to better performance?

(Q3) **LAS vs UAS.** Isn't the UAS (unlabelled) metric more relevant for model comparison as opposed to LAS since the labels are anyway not predicted by the proposed method? I thought it would make more sense to have that result in the main paper compared to having it in the appendix. Also, it seems surprising that LAS with only 10 training examples is so high. Do 10 examples even have enough variations with all labeled arcs? It seems a bit odd.


Typos:
Line 301: objetive -> objective

**Limitations:**

Limitations are discussed in the final section.

**Strengths And Weaknesses:**



### Strengths


(S1) **Interesting Method and theoretical results.** The adaption of the DRO framework for tree-structured prediction is quite interesting. The issues with prior approaches are well described and show a natural fit for the proposed approach. Provable methods for structured prediction are relatively scarce due to various reasons and this paper seems to take a step in that direction.


(S2) **Efficiency and convergence.** The designed algorithms seem to be reasonably efficient and fast convergence makes it appealing from a practical standpoint.


(S3) **Writing.** The paper is well written, the motivations are clearly outlined and the arguments are well-presented.


### Weaknesses


(W1) **Incremental improvement.** Compared to the baseline (BiAF), the proposed method achieves very small improvements across all tasks even in the low training data setting.

(W2) **Reliance on prior representations.** Although representation learning is not the main goal of the paper, the disconnect with the representation learning model puts it in a position of disadvantage. In experiments, they first train the baseline model (Roberta+BiAF) on the training data and then use its representation (assuming - prefinal layer) as features for the proposed model. Compared to end-to-end models, the proposed method could perform worse due to the lack of proper features.


**General Comment.** Overall, I think it is an interesting paper. Despite weak practical gains, the work could be helpful to other researchers in the community working on such structured prediction problems.

---

> ### Author Response · Authors · 2022-08-02
> **Response to Reviewer sgAv**
>
> Thanks for the positive comment.
>
> **Q: Low resources languages.**
>
> A: Thanks for this good suggestion. We will consider adding experiments results in our revision on a low-resource language dataset in UD without using BERT embeddings.
>
> **Q: Representation learning. Is it feasible to design an algorithm with the proposed objective that also updates the feature representation (using some different optimization method)? In that, I guess convergence guarantees and generalization bounds may not hold but may lead to better performance?**
>
> A: Our method can be easily adapted to an end-to-end automatic differentiation framework. Incorporating automatic representation learning into our method is indeed highly desired because of its practical value in applications. We omitted the discussion of this topic in our initial submission due to space limits, but will include it in our revision if extra space is available.
> Although any representation learning model can be adopted, we focus on discussing the most popular one nowadays, the neural network model with end-to-end learning and automatic differentiation. We show how to make use of our DRO method as the final loss layer in a neural network model. A network for supervised learning typically has a linear classification layer in the end without activation. Assume the penultimate layer outputs $\boldsymbol{\Phi}(\boldsymbol{x}) \in \mathbb{R}^{k \times d}$ for input $\boldsymbol{x}$, the last layer will typically output  $\boldsymbol{\psi}(\boldsymbol{x}) := \boldsymbol{\Phi}(\boldsymbol{x}) \boldsymbol{\theta} \in \mathbb{R}^{k}$ for some $\boldsymbol{\theta} \in \mathbb{R}^{d}$. $\boldsymbol{\psi}(\boldsymbol{x})$ is sometimes called logits and yields probability distribution with a softmax layer. For example, in univariate classification, $k$ is the number of labels. In dependency parsing, $k = n^2$ with $n$ being the number of tokens in the input sentence $\boldsymbol{x}$. Given $b$-dimensional token-wise embeddings before the penultimate layer, the biaffine layer in BiAF yields $\boldsymbol{\Phi}(\boldsymbol{x}) \in \mathbb{R}^{n^2 \times b^2}$ equivalently ($b^2$-dimensional feature vector for each arc). Thus $\boldsymbol{\psi}(\boldsymbol{x}) \in \mathbb{R}^{n^2}$ is the logits for all the arcs. Note that $\boldsymbol{\theta}$ in our formulation is naturally equivalent to the parameters of the aforementioned last linear layer. Therefore having $\boldsymbol{\psi}(\boldsymbol{x})$ is sufficient for us to compute $\mathbb{P}^*_{Y|\boldsymbol{x}}$ and $\mathbb{Q}^*_{Y|\boldsymbol{x}}$. In this way, our method is the loss layer without learnable parameters, which backpropagates the sub-derivative of the objective with respect to $\boldsymbol{\psi}(\boldsymbol{x})$ to the linear classification layer: $\partial{\text{Obj}}/\partial{\boldsymbol{\psi}(\boldsymbol{x})} \triangleq \sum_{i = 1}^{B}  {\mathbf q}^{(i)*} - {\mathbf p}_{\text{emp}}^{(i)}$, where $B$ is the batch size. Recall $\mathbf{q}$ and ${\mathbf p}_\text{emp}$ are the probability vectors for $\mathbb{Q}$ and $\mathbb{P}^{\text{emp}}$ respectively. The sub-derivative of the regularization term with respect to $\boldsymbol{\theta}$ is added to the classification layer. Although losing global convergent and provable generalization guarantees, we are now able to take advantage of automatic differentiation and focus on solving the minimax problem given $\boldsymbol{\psi}(\boldsymbol{x})$ and groundtruth $\boldsymbol{y}$ for training. Since the computational bottleneck lies in computing $\boldsymbol{\Phi}(\boldsymbol{x}) \boldsymbol{\theta}$ while GPU acceleration now does it for us, the overhead of computing the adversarial loss should not be much higher than that of computing the cross-entropy loss.
>
>
> **Q: LAS vs UAS. I thought it would make more sense to have that result in the main paper compared to having it in the appendix. Also, it seems surprising that LAS with only 10 training examples is so high. Do 10 examples even have enough variations with all labeled arcs? It seems a bit odd.**
>
> A: We agree that UAS is more relevant in our setting. We will change LAS to UAS or include both in our revision. Since BiAF trains a classifier to predict relational labels independently for each arc, the baseline is also trained to predict the unlabeled tree. The surprising result of high LAS with only 10 training examples is possibly because (1) the backbone and biaffine layer were trained together on the whole training set that contains our training subset; (2) the powerful backbone network along with BERT embeddings has learned to represent the real-world data near a low-dimensional manifold in an easily linearly separable way. Intuitively, BiAF was trained to select the correct head node for each node, which is easier than selecting the correct tree. 10 examples with an average length of 20 would yield 10 * 20 * 20 balanced instances for the classifier to discriminate between.

---

### Official Review · Reviewer_8tu8 · 2022-07-12

**Rating:** 7
**Confidence:** 1
**Soundness:** 4 excellent
**Presentation:** 4 excellent
**Contribution:** 4 excellent

**Summary:**

The authors propose a distributionally robust optimization (DRO) framework for structured prediction over trees, where the ambiguity set is given by a maximum feature moment difference to the empirical distribution. They derive an unconstrained dual problem and establish a worst-case excess risk bound for the optimal parameter. They discuss strategies for efficient (approximate) computation of this parameter, with particular emphasis on projection onto the arborescence polytope. The method is illustrated on three dependency parsing tasks, where it performs competitively with a neural dependency parser, with consistent outperformance in low-data settings.

**Questions:**

- In Section 6, the authors argue that training BiAF on an ERM objective means that the pretrained features may be suboptimal for the DRO objective. But doesn't this also suggest that BiAF will suffer from test evaluation in terms of attachment score (i.e. Hamming loss), while this is being optimized directly in the DRO method?

- In Table 1 the authors compare their method to BiAF in terms of compute time per gradient descent step. How do the total training times compare?

**Limitations:**

Some tradeoffs and assumptions are discussed at the end of Section 7. The authors acknowledge that computation time may be a bottleneck and address tradeoffs in computation between the Marginal and Stochastic / Game implementations; they point out (fairly) that for the increase in computational cost versus the BiAF baseline they are able to offer a guarantee of distributional robustness.



**Strengths And Weaknesses:**

Strengths.

- This is a very well-written paper. The writing is direct, precise, and information-dense. The exposition makes extensive reference to relevant work, providing rich context and highlighting the gaps that the authors aim to fill.

- The contribution appears novel and and is thoroughly developed in its theoretical, computational, and empirical aspects. There are significant technical contributions in terms of a novel DRO problem formulation and its analysis, in terms of the statistical properties of the associated estimator, and in terms of considerations for its efficient computation. The experiments compare the method to a strong baseline, quantify uncertainty in the result, and consider both accuracy metrics and computational costs.

Weaknesses.

- Except on the UD dataset, the empirical results are somewhat modest outside of two very-low-data settings ($m=10$ and $m=50$).

---

> ### Author Response · Authors · 2022-08-02
> **Response to Reviewer 8tu8**
>
> Thanks for the positive comment.
>
> **Q: Except on the UD dataset, the empirical results are somewhat modest outside of two very-low-data settings (m=10 and m=50).**
>
> A: **Modest Improvement**. First of all, we try our best to make fair comparisons in the experiments. We adopted the pretrained embeddings of the state-of-the-art neural parser to be closer to Fisher consistency’s ideal assumption of optimizing over all measurable functions. The pretrained features produced by complicated non-linear models are therefore good enough for even a simple linear classifier to work. Note that both the baseline and our method have the same number of learnable parameters and can be considered linear classifiers. In this way, it is not surprising the improvement seems to be modest, even though we have tested the statistical significance. What’s more, given that the linear classification layer of the baseline was trained together with the backbone during pretraining, and the adopted datasets are all highly competitive dependency parsing benchmarks that have been attacked for years, we believe that a 0.5% improvement in UAS/LAS is sufficient to demonstrate the benefit of replacing the training and inference steps with our method without changing the number of model parameters. We expect a larger margin of improvement by leveraging end-to-end learning of the features and our model parameters altogether.
>
> **Q: In Section 6, the authors argue that training BiAF on an ERM objective means that the pretrained features may be suboptimal for the DRO objective. But doesn't this also suggest that BiAF will suffer from test evaluation in terms of attachment score (i.e. Hamming loss), while this is being optimized directly in the DRO method?**
>
> A: A natural setting for comparing all methods fairly is when each is given the same input feature. Being able to consider the test evaluation conditions in a Fisher consistent manner during training is one of the advantages of the DRO method. And our experiment results show the importance of aligning the training objective with the test objective. BiAF and other state-of-the-art neural parsers all seek to minimize the log-likelihood with different normalization methods, e.g., global, head selection or arc-wise. However, none of them adopts the risk objective suggested by [1], probably because for automatic differentiation methods, the objective is non-differentiable, piece-wise constant, thus difficult to optimize, where smoothing heuristics are necessary to make it work [2][3]. Incorporating test-loss-guided risk minimization into neural network training to enhance its empirical performance is an interesting future direction to investigate.
>
> **Q: In Table 1 the authors compare their method to BiAF in terms of compute time per gradient descent step. How do the total training times compare?**
>
> A: In our experiments, we observed that all the methods take about 150-300 steps to get to the optimal performance on the validation set. So for 200 training samples, the marginal approach is as fast as BiAF while the stochastic approach is 7 times slower than BiAF. Note that BiAF only involves computing a linear combination of features and a summation for backpropagation whereas the stochastic and game DRO methods have to solve a saddle-point problem with iterative methods per gradient step. However, if representation learning is enabled, the computational cost is likely to be dominated by backpropagation in the backbone network. In this regard, the additional cost of replacing the smooth surrogate loss with our method is not significant.
>
>
> [1] Stoyanov, Veselin, and Jason Eisner. "Minimum-risk training of approximate CRF-based NLP systems." In Proceedings of the 2012 Conference of the North American Chapter of the Association for Computational Linguistics: Human Language Technologies, pp. 120-130. 2012.
>
> [2] Mensch, Arthur, and Mathieu Blondel. "Differentiable dynamic programming for structured prediction and attention." In International Conference on Machine Learning, pp. 3462-3471. PMLR, 2018.
>
> [3] Gormley, Matthew R., Mark Dredze, and Jason Eisner. "Approximation-aware dependency parsing by belief propagation." Transactions of the Association for Computational Linguistics 3 (2015): 489-501.

---

### Meta-Review · Area_Chair_DQ2n · 2022-08-28

**Recommendation:** Accept
**Confidence:** Certain

**Metareview:**

This paper presents theoretical results for structured prediction over trees and empirical results on three syntactic dependency parsing datasets. All reviewers agree that the paper is well written and novel. The empirical gains are not huge, but the comparisons are done in a thorough and fair way. All reviewers suggest acceptance (even though some reviewers have low confidence) and the meta reviewer agrees as well.

**Award:**

No

---

### Decision · Program_Chairs · 2022-09-14

Accept